# Pave Your Own Path: Graph Gradual Domain Adaptation on Fused Gromov-Wasserstein Geodesics

**Zhichen Zeng**                                                                 *zhichenz@illinois.edu*
*University of Illinois Urbana-Champaign*

**Ruizhong Qiu**                                                                        *rq5@illinois.edu*
*University of Illinois Urbana-Champaign*

**Wenxuan Bao**                                                                      *wbao4@illinois.edu*
*University of Illinois Urbana-Champaign*

**Tianxin Wei**                                                                      *twei10@illinois.edu*
*University of Illinois Urbana-Champaign*

**Xiao Lin**                                                                        *xiaol13@illinois.edu*
*University of Illinois Urbana-Champaign*

**Yuchen Yan**                                                                     *yucheny5@illinois.edu*
*University of Illinois Urbana-Champaign*

**Tarek F. Abdelzaher**                                                               *zaher@illinois.edu*
*University of Illinois Urbana-Champaign*

**Jiawei Han**                                                                         *hanj@illinois.edu*
*University of Illinois Urbana-Champaign*

**Hanghang Tong**                                                                    *htong@illinois.edu*
*University of Illinois Urbana-Champaign*

**Reviewed on OpenReview:** *https://openreview.net/forum?id=dTPBqTKGPs*

## Abstract

Graph neural networks, despite their impressive performance, are highly vulnerable to distribution shifts on graphs. Existing graph domain adaptation (graph DA) methods often implicitly assume a *mild* shift between source and target graphs, limiting their applicability to real-world scenarios with *large* shifts. Gradual domain adaptation (GDA) has emerged as a promising approach for addressing large shifts by gradually adapting the source model to the target domain via a path of unlabeled intermediate domains. Existing GDA methods exclusively focus on independent and identically distributed (IID) data with a predefined path, leaving their extension to *non-IID graphs without a given path* an open challenge. To bridge this gap, we present GADGET, the first GDA framework for non-IID graph data. First (*theoretical foundation*), the Fused Gromov-Wasserstein (FGW) distance is adopted as the domain discrepancy for non-IID graphs, based on which, we derive an error bound on node, edge and graph-level tasks, showing that the target domain error is proportional to the length of the path. Second (*optimal path*), guided by the error bound, we identify the FGW geodesic as the optimal path, which can be efficiently generated by our proposed algorithm. The generated path can be seamlessly integrated with existing graph DA methods to handle large shifts on graphs, improving state-of-the-art graph DA methods by up to 6.8% in accuracy on real-world datasets.

# 1   Introduction

In the era of big data and AI, graphs have emerged as a powerful tool for modeling relational data. Graph neural networks (GNNs) have achieved remarkable success in numerous graph learning tasks such as graph classification Xu et al. (2018), node classification Kipf & Welling (2017), and link prediction Zhang & Chen (2018). Their superior performance largely relies on the fundamental assumption that training and test graphs are identically distributed, whereas the large distribution shifts on real-world graphs significantly undermine GNN performance Li et al. (2022).

To address this issue, graph domain adaptation (graph DA) aims to adapt the trained source GNN model to a test target graph Wu et al. (2023); Liu et al. (2023a). Promising as it might be, existing graph DA methods follow a fundamental assumption that the source and target graphs bear *mild* shifts, while real-world graphs could suffer from *large* shifts in both node attributes and graph structure Hendrycks et al. (2021); Shi et al. (2024). For example, user profiles are likely to vary from different research platforms (e.g., ACM and DBLP), resulting in attribute shifts on citation networks. In addition, while Instagram users are prone to connect with close friends, users tend to connect to business partners on LinkedIn, leading to structure shifts on social networks. To handle large shifts, gradual domain adaptation (GDA) has emerged as a promising approach Kumar et al. (2020); Wang et al. (2022); He et al. (2023). The key idea is to gradually adapt the source model to the target domain via a path of unlabeled intermediate domains, such that the mild shifts between successive domains are easy to handle. Existing GDA approaches exclusively focus on independent and identically distributed (IID) data, e.g., images, with a predefined path Kumar et al. (2020); Wang et al. (2022), however, the extension of GDA to non-IID graphs without a predefined path remains an open challenge. Therefore, a question naturally arises:

*How to perform GDA on graphs such that large graph shifts can be effectively handled?*

**Contributions.** In this work, we focus on the unsupervised graph DA and propose GADGET, the first GDA framework for non-IID graphs with large shifts. An illustration of GADGET is shown in Figure 1. While direct graph DA fails when facing large shifts (Figure 1(a)), GADGET gradually adapts the GNN model via unlabeled intermediate graphs based on self-training (Figure 1(b)), achieving significant improvement on graph DA methods on real-world graphs (Figure 1(c)). Specifically, to measure the domain discrepancy between non-IID graphs, we adopt the prevalent Fused Gromov-Wasserstein (FGW) distance Titouan et al. (2019) considering both node attributes and connectivity, such that the node dependency, i.e., non-IID property, of graphs can be modeled. Afterwards (*theoretical foundation*), we derive an error bound for graph GDA, revealing the close relationship between the target domain error and the length of the path. Furthermore (*optimal path*), based on the established error bound, we prove that the FGW geodesic minimizing the path length provides the optimal path for graph GDA. To address the lack of path in graph learning tasks, we propose a fast algorithm to generate intermediate graphs on the FGW geodesics, which can be seamlessly integrated with various graph DA baselines to handle large graph shifts. Finally (*empirical evaluation*), we carry out experiments on node-level classification, and the results demonstrate the effectiveness of our proposed GADGET, significantly improving graph DA methods by up to 6.8% in classification accuracy.

# 2   Related Works

**Graph Domain Adaptation.** Graph DA transfers knowledge between graphs with different distributions and can be broadly categorized into data and model adaptation. For data adaptation, shifts between source and target graphs are mitigated via deep transformation Jin et al. (2023); Sui et al. (2023), edge re-weighting Liu et al. (2023a) and graph alignmentLiu et al. (2024a). For model adaptation, various general domain discrepancies, e.g., MMD Gretton et al. (2012) and CORAL Sun et al. (2016), and graph domain discrepancies Zhu et al. (2021); Wu et al. (2023); You et al. (2023), are proposed to align the source and target distributions. In addition, adversarial approaches Dai et al. (2022); Zhang et al. (2019) learn domain-adaptive embeddings that are robust to domain shifts. However, existing graph DA methods only handle mild shifts between source and target, limiting their application to real-world large shifts.

**Gradual Domain Adaptation.** GDA tackles large domain shifts by leveraging gradual transitions along intermediate domains. GDA is first studied in Kumar et al. (2020), where the self-training paradigm and

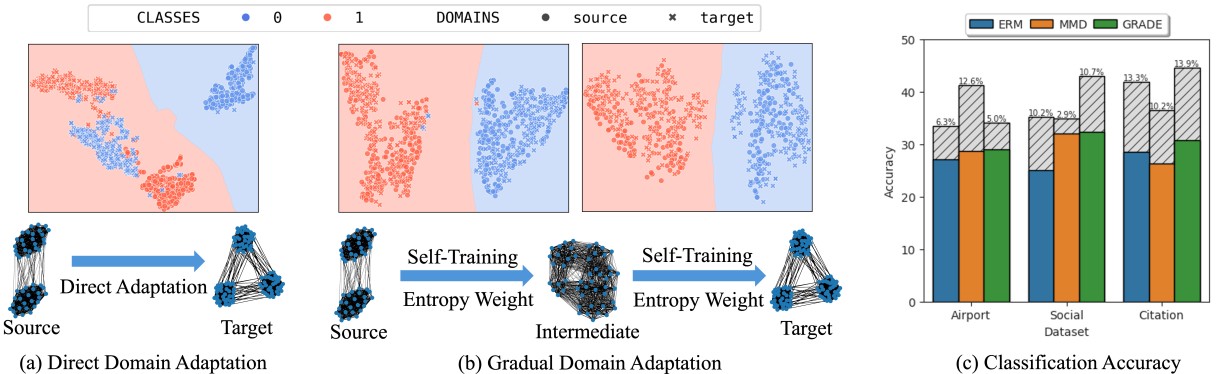

Figure 1: An illustration of graph GDA. Figures (a-b) show the node embeddings, whose colors (blue and red) indicate classes and shapes ($\bullet$ and $\times$) indicate domains, and the decision boundary. (a): Direct adaptation fails when facing large shifts as all target nodes in class 0 ($\times$) are misclassified. (b): Gradual adaptation successfully handles large shift by decomposing it into intermediate domains on the FGW geodesics with mild shifts, where all target nodes in class 0 ($\times$) are correctly separated from those in class 1 ($\times$). (c): Bars w/ and w/o hatches show the performance of direct adaptation and GDA, respectively. Number over bars are the absolute improvement on accuracy. Our proposed GADGET significantly improves various graph DA methods on real-world datasets.

its error bound, are proposed. More in-depth theoretical insights Wang et al. (2022) identify optimal paths, achieving trade-offs between efficiency and effectiveness. More recent studies generalize GDA to scenarios without well-defined intermediate domain by either selecting from a candidate pool Chen & Chao (2021) or generating from scratch He et al. (2023). However, existing GDA methods exclusively focus on IID data, whereas the extension to non-IID graph data is largely un-explored.

## 3 Preliminaries

In this section, we first introduce the notations in Section 3.1, based on which, preliminaries on the FGW space and graph DA are introduced in Sections 3.2 and 3.3, respectively.

### 3.1 Notations

We use bold uppercase letters for matrices (e.g., $\boldsymbol{A}$), bold lowercase letters for vectors (e.g., $\boldsymbol{s}$), calligraphic letters for sets (e.g., $\mathcal{G}$), and lowercase letters for scalars (e.g., $\alpha$). The element $(i,j)$ of a matrix $\boldsymbol{A}$ is denoted as $\boldsymbol{A}(i,j)$. The transpose of $\boldsymbol{A}$ is denoted by the superscript $\top$ (e.g., $\boldsymbol{A}^\top$).

We use $\mathcal{X}$ for feature space and $\mathcal{Y}$ for prediction space, with their respective norms as $\|\cdot\|_{\mathcal{X}}$ and $\|\cdot\|_{\mathcal{Y}}$. A graph $\mathcal{G} = (\mathcal{V}, \boldsymbol{A}, \boldsymbol{X})$ has node set $\mathcal{V}$, adjacency matrix $\boldsymbol{A} \in \mathbb{R}^{|\mathcal{V}| \times |\mathcal{V}|}$ and node feature matrix $\boldsymbol{X} \in \mathcal{X}^{|\mathcal{V}|}$. Let $\mathfrak{G}$ denote the space of all graphs, a GNN is a function $f : \mathfrak{G} \to \mathcal{Y}^{|\mathcal{V}|}$ mapping a graph $\mathcal{G} \in \mathfrak{G}$ to the prediction space $\mathcal{Y}$. We denote the source graph by $\mathcal{G}_0$ and the target graph by $\mathcal{G}_1$. We use subscripts $n, e, g$ to denote node-level, edge-level and graph-level tasks, respectively.

The simplex histogram with $n$ bins is denoted as $\Delta_n = \{\boldsymbol{\mu} \in \mathbb{R}_n^+ | \sum_{i=1}^n \boldsymbol{\mu}(i) = 1\}$. We denote the probabilistic coupling as $\Pi(\cdot, \cdot)$, and the inner product as $\langle \cdot, \cdot \rangle$. We use $\delta_x$ to denote the Dirac measure in $x$. For simplicity, we denote the set of positive integers no greater than $n$ as $\mathbb{N}_{\leq n}^+$.

### 3.2 Fused Gromov–Wasserstein (FGW) Space

The FGW distance serves as a powerful measure for non-IID graph data by considering both node attributes and connectivity. Formally, the FGW distance can be defined as follows.

**Definition 1** (FGW distance: Peyré et al. (2016; 2019); Titouan et al. (2019))**.** Given two graphs $\mathcal{G}_0, \mathcal{G}_1$ represented by probability measures $\boldsymbol{\mu}_0 = \sum_{i=1}^{|\mathcal{V}_0|} h_i \delta_{(v_i, \boldsymbol{X}_0(v_i))}, \boldsymbol{\mu}_1 = \sum_{j=1}^{|\mathcal{V}_1|} g_j \delta_{(u_j, \boldsymbol{X}_1(u_j))}$, where $h \in \Delta_{|\mathcal{V}_0|}, g \in$

$\Delta_{|\mathcal{V}_1|}$ are histograms, a cross-graph matrix $\boldsymbol{M} \in \mathbb{R}^{|\mathcal{V}_0| \times |\mathcal{V}_1|}$ measuring cross-graph node distances based on attributes, and two intra-graph matrices $\boldsymbol{C}_0 \in \mathbb{R}^{|\mathcal{V}_0| \times |\mathcal{V}_0|}, \boldsymbol{C}_1 \in \mathbb{R}^{|\mathcal{V}_1| \times |\mathcal{V}_1|}$ measuring intra-graph node similarity based on graph structure, the FGW distance $d_{\text{FGW};q,\alpha}(\mathcal{G}_0, \mathcal{G}_1)$ is defined as:

$$
\begin{aligned}
d_{\text{FGW};q,\alpha}(\mathcal{G}_1, \mathcal{G}_2) &= \min_{\boldsymbol{S} \in \Pi(\boldsymbol{\mu}_0, \boldsymbol{\mu}_1)} \left( \varepsilon_{\mathcal{G}_1, \mathcal{G}_2}(\boldsymbol{S}) \right)^{\frac{1}{q}}, \text{ where} \\
\varepsilon_{\mathcal{G}_1, \mathcal{G}_2}(\boldsymbol{S}) &= \sum_{\substack{u \in \mathcal{V}_0 \\ v \in \mathcal{V}_1}} (1-\alpha) \boldsymbol{M}(u,v)^q \boldsymbol{S}(u,v) + \sum_{\substack{u,u' \in \mathcal{V}_0 \\ v,v' \in \mathcal{V}_1}} \alpha |\boldsymbol{C}_0(u,u') - \boldsymbol{C}_1(v,v')|^q \boldsymbol{S}(u,v) \boldsymbol{S}(u',v'),
\end{aligned} \tag{1}
$$

where $q$ and $\alpha$ are the order and weight parameters of the FGW distance, respectively.

Intuitively, the FGW distance calculates the optimal matching $\boldsymbol{S}$ between two graphs in terms of both attribute distance $\boldsymbol{M}$ and node connectivity $\boldsymbol{C}_0, \boldsymbol{C}_1$. Following common practice Titouan et al. (2019); Zeng et al. (2024c), we adopt $q = 2$ and use the adjacency matrix $\boldsymbol{A}_i$ as the intra-graph matrices $\boldsymbol{C}_i$. For brevity, we omit the subscripts $q, \alpha$ and use $d_{\text{FGW}}$ to denote $d_{\text{FGW};q,\alpha}$.

Since the FGW distance is only a pseudometric, we follow a standard procedure Howes (2012) to define an induced metric $d_{\text{FGW}}^*$. We start with the FGW equivalence class defined as follows.

**Definition 2** (FGW equivalence class)**.** Given two graphs $\mathcal{G}_0, \mathcal{G}_1$, the FGW equivalence relation $\sim$ is defined as $\mathcal{G}_0 \sim \mathcal{G}_1$, iff $d_{\text{FGW}}(\mathcal{G}_0, \mathcal{G}_1) = 0$. The FGW equivalence class w.r.t. $\sim$ is defined as $[\![\mathcal{G}]\!] := \{\mathcal{G}' : \mathcal{G}' \sim \mathcal{G}\}$. The FGW space is defined as $\mathfrak{G}/\sim = \{[\![\mathcal{G}]\!] : \mathcal{G} \in \mathfrak{G}\}$.

Afterwards, the induced metric $d_{\text{FGW}}^*$ is defined by $d_{\text{FGW}}^*([\![\mathcal{G}_0]\!], [\![\mathcal{G}_1]\!]) = d_{\text{FGW}}(\mathcal{G}_0, \mathcal{G}_1)$, which measures the distance between two FGW equivalence classes. The FGW geodesics is defined as follows

**Definition 3** (FGW geodesic)**.** A curve $\gamma : [0,1] \to \mathfrak{G}/\sim$ is an FGW geodesic from $[\![\mathcal{G}_0]\!]$ to $[\![\mathcal{G}_1]\!]$ iff $\gamma(0) = [\![\mathcal{G}_0]\!]$, $\gamma(1) = [\![\mathcal{G}_1]\!]$, and for every $\lambda_0, \lambda_1 \in [0,1]$,

$$
d_{\text{FGW}}^*(\gamma(\lambda_0), \gamma(\lambda_1)) = |\lambda_0 - \lambda_1| \cdot d_{\text{FGW}}^*([\![\mathcal{G}_0]\!], [\![\mathcal{G}_1]\!]).
$$

Intuitively, the FGW geodesic is the shortest line directly linking the source and target graph. To simplify notation, we use $[\![\mathcal{G}]\!]$ and $\mathcal{G}$ interchangeably for the rest of the paper.

### 3.3 Unsupervised Graph Domain Adaptation

Unsupervised graph DA aims to adapt a GNN model trained on a labeled source graph to an unlabeled target graph, which can be formally defined as follows.

**Definition 4** (Unsupervised graph DA)**.** Given a source graph $\mathcal{G}_0$ with labels $\boldsymbol{Y}_0$, where $\boldsymbol{Y}_0 \in \mathcal{Y}_n^{|\mathcal{V}_0|}$ for node-level task, $\boldsymbol{Y}_0 \in \mathcal{Y}_e^{|\mathcal{V}_0| \times |\mathcal{V}_0|}$ for edge-level tasks and $\boldsymbol{Y}_0 \in \mathcal{Y}_g$ for graph-level tasks, and a target graph $\mathcal{G}_1$. Unsupervised graph DA aims to train a model $f$ using the labeled source graph $(\mathcal{G}_0, \boldsymbol{Y}_0)$ and the unlabeled target graph $\mathcal{G}_1$ to accurately predict target labels $\widehat{\boldsymbol{Y}}_1 = f(\mathcal{G}_1)$.

However, existing graph DA methods fundamentally assume mild shifts between source and target graphs. To handle large shifts, we introduce the idea of GDA to graph DA, which gradually adapts a source GNN to the target graph via a series of sequentially generated graphs.

## 4 Theoretical Foundation

In this section, we present the theoretical foundation of graph GDA. The problem is formulated in Section 4.1. We establish the error bound in Section 4.3 and derive the optimal path in Section 4.4.

### 4.1 Problem Setup

To formulate the graph GDA problem, we first define the path for graph GDA as follows.

**Definition 5** (Path). A path between the source graph $\mathcal{G}_0$ and target graph $\mathcal{G}_1$ is defined as $\mathcal{H} = (\mathcal{H}_0, \mathcal{H}_1, ..., \mathcal{H}_T)$, where $\mathcal{H}_0 = \mathcal{G}_0$ and $\mathcal{H}_T = \mathcal{G}_1$.

In general, for a $T$-stage graph GDA, given the model $f_{t-1}$ at stage $t-1$ and the successive graph $\mathcal{H}_t$ at stage $t$, self-training paradigm trains the successive model $f_t$ based on the pseudo-labels $f_{t-1}(\mathcal{H}_t)$. Formally, graph GDA can be defined as follows.

**Definition 6** (Graph gradual domain adaptation). Given a source graph $\mathcal{G}_0$ with label $\boldsymbol{Y}_0$, and a target graph $\mathcal{G}_1$. Graph GDA (1) finds a path $\mathcal{H}$ with $\mathcal{H}_0 = \mathcal{G}_0, \mathcal{H}_T = \mathcal{G}_1$, and (2) gradually adapts the source model to the target graph via self-training, that is:

$$f_t := \arg\min_{f_t} \ell\left(f_t(\mathcal{H}_t), f_{t-1}(\mathcal{H}_t)\right), \forall t = 1, 2, ..., T,$$

where $\ell$ is the loss function and $f_{t-1}(\mathcal{H}_t)$ is the pseudo-label for the $t$-th graph $\mathcal{H}_t$ given by the previous model $f_{t-1}$. Graph GDA aims to minimize the target error between the prediction $f_T(\mathcal{G}_1)$ and the groundtruth label $\boldsymbol{Y}_1$.

Note that we consider a more general self-training paradigm compared to Empirical Risk Minimization (ERM) Kumar et al. (2020) and do not pose specific constraints on the loss function $\ell$. That is to say, *our proposed framework is compatible with various graph DA baselines with different adaptation losses.*

**Definition 7** (Graph convolution). For any graph $\mathcal{G} = (\mathcal{V}, \boldsymbol{A}, \boldsymbol{X})$, the graph convolution operation gcn for any node $u \in \mathcal{V}$ depends only on node pair information $\mathcal{N}_\mathcal{G}(u) := \{\boldsymbol{A}(u, v), \boldsymbol{X}(v)\}_{v \in \mathcal{V}}$, that is

$$\text{gcn}(\mathcal{G})_u := \text{gcn}(\mathcal{N}_\mathcal{G}(u)) := \text{gcn}(\{\boldsymbol{A}(u, v), \boldsymbol{X}(v)\}_{v \in \mathcal{V}}), \forall u \in \mathcal{V}.$$

A GNN layer $g^{(i)}$ is a composition of graph convolution gcn, linear transformation and ReLU activation

$$g^{(i)} = \text{ReLU} \circ \text{Linear} \circ \text{gcn}^{(i)}. \tag{2}$$

We further define node-level, edge-level and graph-level tasks as follows

**Definition 8** (Node-level task). A GNN model is a composition of graph convolutions $g^{(i)}$, i.e., $f_n = g^{(L)} \circ ... \circ g^{(1)}$. For each node $u \in \mathcal{V}$, the node-level loss is defined by $\epsilon_n(f_n(\mathcal{G})_u)$, where the groundtruth label $\boldsymbol{Y}(u)$ is omitted for brevity. The overall node-level loss of a GNN $f_n$ on a graph $\mathcal{G}$ can be defined as

$$\xi_n(f_n, \mathcal{G}) := \frac{1}{|\mathcal{V}|} \sum_{u \in \mathcal{V}} \epsilon_n(f_n(\mathcal{G})_u).$$

**Definition 9** (Edge-level task). A GNN model is a composition of graph convolutions $g^{(i)}$ and a pairwise aggregation function $\phi$, i.e., $f_e = \phi \circ g^{(L)} \circ ... \circ g^{(1)} = \phi \circ f_n$. The aggregation function $\phi$ turns the embeddings of two nodes $f_n(\mathcal{G})_u, f_n(\mathcal{G})_{u'}$ into an edge embedding $f_e(\mathcal{G})_{(u,u')} = \phi(f_n(\mathcal{G})_u, f_n(\mathcal{G})_{u'})$. For each edge $(u, u') \in \mathcal{G}$, the edge-level loss is defined by $\epsilon_e\left(f_e(\mathcal{G})_{(u,u')}\right)$, where the groundtruth label $\boldsymbol{Y}((u, u'))$ is omitted for brevity. The overall edge-level loss of a GNN $f_e$ on a graph $\mathcal{G}$ can be defined as

$$\xi_e(f_e, \mathcal{G}) = \frac{1}{|\mathcal{V}|^2} \sum_{u, u' \in \mathcal{V}} \epsilon_e\left(f_e(\mathcal{G})_{(u,u')}\right).$$

**Definition 10** (Graph-level task). A GNN model is a composition of graph convolutions $g^{(i)}$ and a pooling function $r$, i.e., $f_g = r \circ g^{(L)} \circ ... \circ g^{(1)}$. The pooling function $r$ turns the embeddings of all nodes $f_n(\mathcal{G})$ into a graph embedding $f_g(\mathcal{G}) = r(f_n(\mathcal{G}))$. The overall graph-level loss of a GNN $f_g$ on a graph $\mathcal{G}$ can be defined as $\xi_g(f_g, \mathcal{G})$, where the groundtruth label $y(\mathcal{G})$ is omitted for brevity.

## 4.2 Assumptions

To capture the non-IID nature, i.e., node dependency, of graphs, we adopt the FGW distance Titouan et al. (2019) in equation 1 as the domain discrepancy, measuring the graph distance in terms of both node attributes $\boldsymbol{X}$ and node connectivity $\boldsymbol{A}$. We make several assumptions following previous works on graph DA Zhu et al. (2021); Bao et al. (2024) and GDA Kumar et al. (2020); Wang et al. (2022).

**Assumption 1** (General regularity assumptions). We make several regularity assumptions

**A:** (*Lipschitz continuity of graph convolution*). We assume there exists $C_c > 0$ such that for any nodes $u \in \mathcal{V}_0, v \in \mathcal{V}_1$ we have

$$\|\text{gcn}(\mathcal{G}_0)_u - \text{gcn}(\mathcal{G}_1)_v\|_{\mathcal{X}} \leq C_c \cdot d_W\left(\mathcal{N}_{\mathcal{G}_0}(u), \mathcal{N}_{\mathcal{G}_1}(v)\right),$$
$$\text{where } d_W^q\left(\{(\boldsymbol{A}_0(u, u'), \boldsymbol{X}_0(u'))\}_{u' \in \mathcal{V}_0}, \{(\boldsymbol{A}_1(v, v'), \boldsymbol{X}_1(v'))\}_{v' \in \mathcal{V}_1}\right)$$
$$= \inf_{\boldsymbol{\tau} \in \Pi(\boldsymbol{\mu}_0, \boldsymbol{\mu}_1)} \mathbb{E}_{(u', v') \sim \boldsymbol{\tau}}\left[\alpha|\boldsymbol{A}_0(u, u') - \boldsymbol{A}_1(v, v')|^q + (1 - \alpha)\|\boldsymbol{X}_0(u') - \boldsymbol{X}_1(v')\|_{\mathcal{X}}^q\right].$$

**B:** (*Lipschitz continuity of linear layer*). We assume there exists $C_{\text{lin}} > 0$ such that for any weight matrices $\boldsymbol{W}$ in linear layers $\text{Linear}(\boldsymbol{x}) = \boldsymbol{W}\boldsymbol{x} + \boldsymbol{b}$ we have $\|\boldsymbol{W}\| \leq C_{\text{lin}}$.

Both assumptions ensure the generalization capability and stability of the GNN model. Specifically, Assumption Assumption A enforces smoothness with respect to graph topology: nodes with similar local neighborhoods must yield similar embeddings. Assumption Assumption B requires model parameters to be finite, a standard condition satisfied by regularization.

**Assumption 2** (Task-specific regularity assumptions). We make the following regularity assumptions for different graph learning tasks

**C:** (*Lipschitz continuity of loss functions*). We suppose there exists $C_{f_n} > 0$ for any node-level GNNs $f_{n_i}$, $C_{f_e} > 0$ for any edge-level GNNs $f_{e_i}$, and $C_{f_g}$ for any graph-level GNNs $f_{g_i}$, such that

$$|\epsilon_n(f_{n_0}(\mathcal{G})_u) - \epsilon_n(f_{n_1}(\mathcal{G})_u)| \leq C_{f_n} \cdot \|f_{n_0}(\mathcal{N}_{\mathcal{G}}(u)) - f_{n_1}(\mathcal{N}_{\mathcal{G}}(u))\|_{\mathcal{Y}_n}, \tag{3}$$

$$|\epsilon_e(f_{e_0}(\mathcal{G})_{(u, u')}) - \epsilon_e(f_{e_1}(\mathcal{G})_{(u, u')})| \leq C_{f_e} \cdot \|f_{e_0}(\mathcal{G})_{(u, u')} - f_{e_1}(\mathcal{G})_{(u, u')}\|_{\mathcal{Y}_e}, \tag{4}$$

$$|\xi_g(f_{g_0}, \mathcal{G}) - \xi_g(f_{g_1}, \mathcal{G})| \leq C_{f_g} \cdot \|f_{g_0}(\mathcal{G}) - f_{g_1}(\mathcal{G})\|_{\mathcal{Y}_g}. \tag{5}$$

**D:** (*Hölder continuity of loss functions*).

We suppose there exists $C_{W_n} > 0$ for any nodes $u \in \mathcal{V}_0, v \in \mathcal{V}_1$; $C_{W_e} > 0$ for any edges $(u, u') \in \mathcal{G}_0, (v, v') \in \mathcal{G}_1$; $C_{W_n} > 0$ for any graphs $\mathcal{G}_1, \mathcal{G}_2$, and $q > 1$, such that

$$|\epsilon_n(f_n(\mathcal{G}_0)_u) - \epsilon_n(f_n(\mathcal{G}_1)_v)| \leq C_{W_n} \cdot \|f_n(\mathcal{G}_0)_u - f_n(\mathcal{G}_1)_v\|_{\mathcal{Y}_n}^q, \tag{6}$$

$$|\epsilon_e(f_e(\mathcal{G}_0)_{(u, u')}) - \epsilon_e(f_e(\mathcal{G}_1)_{(v, v')})| \leq C_{W_e} \cdot \|f_e(\mathcal{G}_0)_{(u, u')} - f_e(\mathcal{G}_1)_{(v, v')}\|_{\mathcal{Y}_e}^q, \tag{7}$$

$$|\xi_g(f_g, \mathcal{G}_0) - \xi_g(f_g, \mathcal{G}_1)| \leq C_{W_g} \cdot \|f_g(\mathcal{G}_0) - f_g(\mathcal{G}_1)\|_{\mathcal{Y}_g}^q. \tag{8}$$

**E:** (*Lipschitz continuity of aggregation function*). We suppose there exists $C_\phi > 0$ such that for any edges $(u, u') \in \mathcal{G}_0, (v, v') \in \mathcal{G}_1$, we have

$$\|\phi(f_n(\mathcal{G}_0)_u, f_n(\mathcal{G}_0)_{u'}) - \phi(f_n(\mathcal{G}_1)_v, f_n(\mathcal{G}_1)_{v'})\|_{\mathcal{Y}_e} \leq C_\phi \cdot (\|f_n(\mathcal{G}_0)_u - f_n(\mathcal{G}_1)_v\|_{\mathcal{X}} + \|f_n(\mathcal{G}_0)_{u'} - f_n(\mathcal{G}_1)_{v'}\|_{\mathcal{X}}). \tag{9}$$

**F:** (*Lipschitz continuity of pooling function*). We suppose there exists $C_r > 0$ such that for any graphs $\mathcal{G}_1, \mathcal{G}_2$ and coupling $\boldsymbol{\pi}$, we have

$$\|r(f_n(\mathcal{G}_0)) - r(f_n(\mathcal{G}_1))\|_{\mathcal{Y}_g} \leq C_r \cdot \mathbb{E}_{(u, v) \sim \boldsymbol{\pi}}\|f_n(\mathcal{G}_0)_u - f_n(\mathcal{G}_1)_v\|_{\mathcal{X}}. \tag{10}$$

Assumption C and Assumption D enforce the smoothness of the loss function with respect to model predictions. Assumption C posits that similar predictions from different models result in similar losses. Assumption D complements this by ensuring that for a fixed model, similar input embeddings lead to similar losses. These conditions are mild and satisfied by standard surrogate losses (e.g., MSE, Cross-Entropy) on bounded domains.

Assumption E and Assumption F guarantee that readout operations preserve embedding proximity. Assumption Assumption E ensures edge-level aggregation remains stable under small perturbations in node embeddings. Similarly, Assumption Assumption F requires the graph pooling function $r$ to preserve local smoothness, ensuring that graphs with aligned node embeddings map to similar graph-level representations.

### 4.3 Error Bound

Under Assumption 1, we analyze the error bound of graph GDA. We first show that any $L$-layer GNN is Hölder continuous w.r.t. the $\beta$-FGW distance, where $\beta = \frac{\alpha\left(1-(C_{\mathrm{c}}C_{\mathrm{lin}}(1-\alpha))^L\right)}{\alpha+(1-\alpha)^L(C_{\mathrm{c}}C_{\mathrm{lin}})^{L-1}(1-C_{\mathrm{c}}C_{\mathrm{lin}})}$.

**Lemma 1** (Hölder continuity). *For any $L$-layer node-level GNN $f_n = \bigcirc_{l=L}^1 g^{(l)}$, edge-level GNN $f_e = \phi \circ \bigcirc_{l=L}^1 g^{(l)}$, and graph-level GNN $f_g = r \circ \bigcirc_{l=L}^1 g^{(l)}$, where $\bigcirc_{l=L}^1 g^{(l)} = g^{(L)} \circ \cdots \circ g^{(1)}$ and $g^{(i)}$ are GNN layers in equation 2. Given a source graph $\mathcal{G}_0$ and a target graph $\mathcal{G}_1$, we have:*

$$|\xi(f, \mathcal{G}_0)) - \xi(f, \mathcal{G}_1)| \le C \cdot d_{\mathrm{FGW};q,\beta}^q(\mathcal{G}_0, \mathcal{G}_1),$$

*where*

$$\beta = \frac{\alpha\left(1-(C_{\mathrm{c}}C_{\mathrm{lin}}(1-\alpha))^L\right)}{\alpha+(1-\alpha)^L(C_{\mathrm{c}}C_{\mathrm{lin}})^{L-1}(1-C_{\mathrm{c}}C_{\mathrm{lin}})}$$

$$C_{\mathrm{gnn}} = C_{\mathrm{c}}C_{\mathrm{lin}}\frac{\alpha+(1-\alpha)(C_{\mathrm{c}}C_{\mathrm{lin}}(1-\alpha))^{L-1} - (C_{\mathrm{c}}C_{\mathrm{lin}}(1-\alpha))^L}{1-C_{\mathrm{c}}C_{\mathrm{lin}}(1-\alpha)}$$

$$C = \begin{cases} C_{\mathrm{W}_n}C_{\mathrm{gnn}}, & \text{for node-level tasks} \\ 2C_{\mathrm{W}_e}C_\phi C_{\mathrm{gnn}}, & \text{for edge-level tasks} \\ C_{\mathrm{W}_g}C_{\mathrm{r}}C_{\mathrm{gnn}}, & \text{for graph-level tasks} \end{cases}$$

The proof can be found in Appendix A. Intuitively, the upper bound of the performance gap between source loss $\xi(f, \mathcal{G}_0)$ and target loss $\xi(f, \mathcal{G}_1)$ is proportional to the FGW distance between the source graph $\mathcal{G}_0$ and target graph $\mathcal{G}_1$. Therefore, GNNs could suffer from significant performance degradation under large shifts.

To alleviate the effects of large shifts, we investigate the effectiveness of applying GDA on graphs, and derive an error bound shown in Theorem 1.

**Theorem 1** (Error bound). *Let $f_0$ denote the source model trained on the source graph $\mathcal{H}_0 = \mathcal{G}_0$. Suppose there are $T-1$ intermediate stages where in the $t$-th stage (for $t = 1, 2, ..., T$), we adapt $f_{t-1}$ to graph $\mathcal{H}_t$ to obtain an adapted $f_t$. If every adaptation step achieves $\|f_{t-1}(\mathcal{H}_t) - f_t(\mathcal{H}_t)\|_\mathcal{Y} \le \delta$ on the corresponding graph $\mathcal{H}_t$, then the final error $\xi(f_T, \mathcal{H}_T)$ on target graph $\mathcal{H}_T = \mathcal{G}_1$ is upper bounded by*

$$\xi(f_T, \mathcal{G}_1) \le \xi(f_0, \mathcal{G}_0) + C_{\mathrm{f}} \cdot \delta T + C \cdot \sum_{t=1}^T d_{\mathrm{FGW};q,\beta}^q(\mathcal{H}_{t-1}, \mathcal{H}_t).$$

*where $C_{\mathrm{f}} = C_{\mathrm{f}_n}$ for node-level task, $C_{\mathrm{f}} = C_{\mathrm{f}_e}$ for edge-level tasks, and $C_{\mathrm{f}} = C_{\mathrm{f}_g}$ for graph-level tasks.*

The proof can be found in Appendix A. In general, the upper bound of the target GNN loss $\xi(f_T, \mathcal{G}_1)$ is determined by three terms, including (1) source GNN loss $\xi(f_0, \mathcal{G}_0)$, (2) the accumulated training error $T\delta$, and (3) the generalization error measured by length of the path $\sum_{t=1}^T d_{\mathrm{FGW}}^q(\mathcal{H}_{t-1}, \mathcal{H}_t)$. In the following subsection, we will analyze which path best benefits the graph GDA process.

### 4.4 Optimal Path

Motivated by Theorem 1, we derive the optimal path that minimizes the error bound in Theorem 2.

**Theorem 2** (Optimal path). *Given a source graph $\mathcal{G}_0$ and a target graph $\mathcal{G}_1$, let $\gamma : [0, 1] \to \mathfrak{G}/\sim$ be an FGW geodesic connecting $\mathcal{G}_0$ and $\mathcal{G}_1$. Then the error bound in Theorem 1 attains its **minimum** when intermediate graphs are $\mathcal{H}_t = \gamma(\frac{t}{T}), \forall t = 0, 1, ..., T$, where we have:*

$$\xi(f_T, \mathcal{G}_1) \le \xi(f_0, \mathcal{G}_0) + C_{\mathrm{f}} \cdot \delta T + \frac{C \cdot d_{\mathrm{FGW};q,\beta}^q(\mathcal{G}_0, \mathcal{G}_1)}{T^{q-1}}.$$

The proof can be found in Appendix A. In general, the key idea is to minimize the path length, whose minimum is achieved by the FGW geodesic between source and target. As a remark, the optimal number $T$

of intermediate steps can be obtained by

$$T \approx \left( \frac{(q-1)C}{C_{\mathrm{f}} \cdot \delta} \right)^{\frac{1}{q}} d_{\mathrm{FGW};q,\beta}(\mathcal{G}_0, \mathcal{G}_1). \tag{11}$$

Intuitively, the number of stages $T$ balances the accumulated training error (the second term on the RHS) and the generalization error (the third term on the RHS). Following Lemma 1, when $C_{\mathrm{W}_n}, C_{\mathrm{W}_e}, C_{\mathrm{W}_g}$ are small, model is robust to domain shifts and the error bound is dominated by the accumulated training error, thus, we expect a smaller $T$ for better performance. On the other hand, when $C_{\mathrm{W}_n}, C_{\mathrm{W}_e}, C_{\mathrm{W}_g}$ are large, model is vulnerable to domain shifts and the error bound is dominated by the generalization error, thus, we expect a larger $T$ to reduce domain shifts, hence achieving better performance.

## 5 Methodology

In this section, we present our proposed GADGET to perform graph GDA on the FGW geodesics. As self-training is highly vulnerable to noisy pseudo labels, we first propose an entropy-based confidence to denoise the noisy labels. Motivated by the theoretical foundation, we introduce a practical algorithm to generate intermediate graphs, which as we prove, reside on the approximated FGW geodesic to best facilitate the graph GDA process.

Motivated by Theorem 2, we generate the FGW geodesic as the optimal path for graph GDA. Previous work Zeng et al. (2024c) generates graphs on the Gromov-Wasserstein geodesic purely based on graph structure via mixup. We generalize such idea to the FGW geodesics to consider both graph structure and node features.

Specifically, given source graph $\mathcal{G}_0 = (\mathcal{V}_0, \boldsymbol{A}_0, \boldsymbol{X}_0)$, target graph $\mathcal{G}_1 = (\mathcal{V}_1, \boldsymbol{A}_1, \boldsymbol{X}_1)$, and their probability distributions $\boldsymbol{\mu}_0, \boldsymbol{\mu}_1$, two transformation matrices $\boldsymbol{P}_0, \boldsymbol{P}_1$ are employed to transform them into well-aligned pairs $\tilde{\mathcal{G}}_0 = (\tilde{\mathcal{V}}_0, \tilde{\boldsymbol{A}}_0, \tilde{\boldsymbol{X}}_0), \tilde{\mathcal{G}}_1 = (\tilde{\mathcal{V}}_1, \tilde{\boldsymbol{A}}_1, \tilde{\boldsymbol{X}}_1)$ with probability distributions $\tilde{\boldsymbol{\mu}}_0, \tilde{\boldsymbol{\mu}}_1$ as follows Zeng et al. (2024c)

$$\begin{aligned}
&\tilde{\boldsymbol{A}}_0 = \boldsymbol{P}_0^\mathsf{T} \boldsymbol{A}_0 \boldsymbol{P}_0, \ \tilde{\boldsymbol{X}}_0 = \boldsymbol{P}_0^\mathsf{T} \boldsymbol{X}_0, \ \tilde{\boldsymbol{\mu}}_0 = \boldsymbol{P}_0^\mathsf{T} \boldsymbol{\mu}_0, \\
&\tilde{\boldsymbol{A}}_1 = \boldsymbol{P}_1^\mathsf{T} \boldsymbol{A}_1 \boldsymbol{P}_1, \ \tilde{\boldsymbol{X}}_1 = \boldsymbol{P}_1^\mathsf{T} \boldsymbol{X}_1, \ \tilde{\boldsymbol{\mu}}_1 = \boldsymbol{P}_1^\mathsf{T} \boldsymbol{\mu}_1, \\
&\text{where } \boldsymbol{P}_0 = \boldsymbol{I}_{|\mathcal{V}_0|} \otimes \boldsymbol{1}_{1 \times |\mathcal{V}_1|}, \ \boldsymbol{P}_1 = \boldsymbol{1}_{1 \times |\mathcal{V}_0|} \otimes \boldsymbol{I}_{|\mathcal{V}_1|}.
\end{aligned} \tag{12}$$

Afterwards, the intermediate graphs $\mathcal{H}_t$ are the interpolations of the well-aligned pairs, that is

$$\mathcal{H}_t := \left( \mathcal{V}_0 \otimes \mathcal{V}_1, \left(1 - \frac{t}{T}\right)\tilde{\boldsymbol{A}}_0 + \frac{t}{T}\tilde{\boldsymbol{A}}_1, \left(1 - \frac{t}{T}\right)\tilde{\boldsymbol{X}}_0 + \frac{t}{T}\tilde{\boldsymbol{X}}_1 \right). \tag{13}$$

With the above transformations, we prove that the intermediate graphs generated by equation 13 are on the FGW geodesics in the following theorem.

**Theorem 3** (FGW geodesic). *Given a source graph $\mathcal{G}_0$ and a target graph $\mathcal{G}_1$, the transformed graphs $\tilde{\mathcal{G}}_0, \tilde{\mathcal{G}}_1$ are in the FGW equivalent class of $\mathcal{G}_0, \mathcal{G}_1$, i.e., $[\![\mathcal{G}_0]\!] = [\![\tilde{\mathcal{G}}_0]\!], [\![\mathcal{G}_1]\!] = [\![\tilde{\mathcal{G}}_1]\!]$. Besides that, the intermediate graphs $\mathcal{H}_t, \forall t = 0, 1, ..., T$, generated by equation 13 are on an FGW geodesic connecting $\mathcal{G}_0$ and $\mathcal{G}_1$.*

According to Theorems 2 and 3, directly applying the generated $\mathcal{H}_t$ best benefits the graph GDA process.

However, practically, the transformations in equation 12 involve computation in the product space, posing great challenges to the scalability to large-scale graphs. For faster computation, we employ an efficient low-rank OT algorithm adapted from Zeng et al. (2024c) to generate intermediate graphs on the FGW geodesics. Specifically, via a change of variable $\boldsymbol{Q}_0 = \boldsymbol{P}_0 \mathrm{diag}(\boldsymbol{g}), \boldsymbol{Q}_1 = \boldsymbol{P}_1 \mathrm{diag}(\boldsymbol{g})$, the transformation matrices $\boldsymbol{P}_0, \boldsymbol{P}_1$ can be obtained by solving the following low-rank OT problem

$$\begin{aligned}
&\arg\min_{\boldsymbol{Q}_0, \boldsymbol{Q}_1, \boldsymbol{g}} (\varepsilon_{\mathcal{G}_0, \mathcal{G}_1} (\boldsymbol{Q}_0^\mathsf{T} \mathrm{diag}(1/\boldsymbol{g}) \boldsymbol{Q}_1))^{\frac{1}{2}}, \\
&\text{s.t. } \boldsymbol{Q}_0 \in \Pi(\boldsymbol{\mu}_1, \boldsymbol{g}), \boldsymbol{Q}_1 \in \Pi(\boldsymbol{\mu}_2, \boldsymbol{g}), \boldsymbol{g} \in \Delta_r,
\end{aligned} \tag{14}$$

where $r$ is the rank of the low-rank OT problem. When $r = |\mathcal{V}_1||\mathcal{V}_2|$, the optimal solution to equation 14 provides the optimal transformation matrices $\boldsymbol{P}_0, \boldsymbol{P}_1$. By reducing the rank of $\boldsymbol{g}$ from $|\mathcal{V}_1||\mathcal{V}_2|$ to a smaller

rank $r$, the low-rank OT problem can be efficiently solved via a mirror descent scheme by iteratively solving the following problem Scetbon et al. (2022); Zeng et al. (2024c):

$$\left(\boldsymbol{Q}_0^{(t+1)}, \boldsymbol{Q}_1^{(t+1)}, \boldsymbol{g}^{(t+1)}\right) = \underset{\boldsymbol{Q}_0, \boldsymbol{Q}_1, \boldsymbol{g}}{\arg\min} \, \mathrm{KL}\left((\boldsymbol{Q}_1, \boldsymbol{Q}_2, \boldsymbol{g}), (\boldsymbol{K}_1^{(t)}, \boldsymbol{K}_2^{(t)}, \boldsymbol{K}_3^{(t)})\right),$$

$$\text{s.t. } \boldsymbol{Q}_0 \in \Pi(\boldsymbol{\mu}_0, \boldsymbol{g}), \; \boldsymbol{Q}_1 \in \Pi(\boldsymbol{\mu}_1, \boldsymbol{g}), \; \boldsymbol{g} \in \Delta_r,$$

$$\text{where} \begin{cases} \boldsymbol{K}_1^{(t)} = \exp\left(\gamma \boldsymbol{B}^{(t)} \boldsymbol{Q}_1^{(t)} \mathrm{diag}(1/\boldsymbol{g}^{(t)})\right) \odot \boldsymbol{Q}_0^{(t)}, \\ \boldsymbol{K}_2^{(t)} = \exp\left(\gamma \boldsymbol{B}^{(t)\top} \boldsymbol{Q}_0^{(t)} \mathrm{diag}(1/\boldsymbol{g}^{(t)})\right) \odot \boldsymbol{Q}_1^{(t)\top}, \\ \boldsymbol{K}_3^{(t)} = \exp\left(-\gamma \mathrm{diag}\left(\boldsymbol{Q}_0^{(t)\top} \boldsymbol{B}^{(t)} \boldsymbol{Q}_1^{(t)}\right)/\boldsymbol{g}^{(t)^2}\right) \odot \boldsymbol{g}^{(t)}, \\ \boldsymbol{B}^{(t)} = -\alpha \boldsymbol{M} + 4(1-\alpha) \boldsymbol{A}_0 \boldsymbol{Q}_0^{(t)} \mathrm{diag}(1/\boldsymbol{g}^{(t)}) \boldsymbol{Q}_1^{(t)\top} \boldsymbol{A}_1. \end{cases}$$

**Remark.** Our path generation algorithm is adapted from Zeng et al. (2024c) but bears subtle difference. First (*space*), the Gromov-Wasserstein (GW) space in Zeng et al. (2024c) only captures graph structure information, but the FGW space considers both node attributes and graph structure information. Secondly (*task*), Zeng et al. (2024c) utilizes the GW geodesics to mixup graphs and their labels for graph-level classification, while GADGET utilizes the FGW geodesic to generate label-free graphs for node-level classification. Thirdly (*label*), Zeng et al. (2024c) utilizes the linear interpolation of graph labels as the pseudo-labels for mixup graphs, requiring information from both ends of the geodesic which is inapplicable for graph GDA, while GADGET utilizes self-training to label the intermediate graphs, relying solely on source information.

**Self-training paradigm.** Self-training is a predominant paradigm for GDA Kumar et al. (2020); Wang et al. (2022), but is known to be vulnerable to noisy pseudo labels Chen et al. (2022a). Such vulnerability may be further exacerbated for GNN models as the noise can propagate Wang et al. (2024); Liu et al. (2022). To alleviate this issue, we utilize an entropy-based confidence to depict the reliability of the pseduo-labels. Given a model output $\widehat{\boldsymbol{y}}_i \in \mathbb{R}^C$, where $C$ is the number of classes, the confidence score $\mathrm{conf}(\widehat{\boldsymbol{y}}_i)$ is calculated by

$$\mathrm{conf}(\widehat{\boldsymbol{y}}_i) := \frac{\max_j \mathrm{ent}(\widehat{\boldsymbol{y}}_j) - \mathrm{ent}(\widehat{\boldsymbol{y}}_i)}{\max_j \mathrm{ent}(\widehat{\boldsymbol{y}}_j) - \min_j \mathrm{ent}(\widehat{\boldsymbol{y}}_j)}, \tag{15}$$

where $\mathrm{ent}(\cdot)$ calculates the entropy of the model prediction. Intuitively, for reliable model outputs, we expect low entropy values and a high confidence scores, and vice versa.

**Error bound under approximation.** Note that the error bound in Theorem 1 applies for the ideal case where intermediate graphs $\mathcal{H}_t$ are on the exact FGW geodesics. The practical algorithm adopts low-rank formulation which may introduce approximation error. We provide the following theorem to quantify the effect of low-rank approximation errors in practical algorithm.

**Theorem 4** (Practical error bound). *For a sequence of intermediate graphs $\tilde{H}_0, \ldots, \tilde{H}_T$ on the approximated FGW geodesics generated by GADGET, performing GDA along the path yield a final error $\xi(f_T, \mathcal{H}_T)$ on target graph $\mathcal{H}_T = \mathcal{G}_1$ upper bounded by*

$$\xi(f_T, \mathcal{G}_1) \leq \text{Original bound} + 4C\delta_{approx} \sum_{t=1}^{T} d_{FGW}(\mathcal{H}_t, \mathcal{H}_{t+1}) + 4CT\delta_{approx}^2$$

*where original bound is the upper bound on the exact geodesics provided in Theorem 1, and $\delta_{approx} = \max_t d_{FGW}(\mathcal{H}_t, \tilde{H}_t)$ is the maximum approximation error between exact geodesic graph $\mathcal{H}_t$ and low-rank approximated graph $\tilde{\mathcal{H}}_t$.*

The proof of Theorem 4 is provided in Appendix A. As detailed in the Appendix, when rank $r \to |\mathcal{V}_1||\mathcal{V}_2|$, i.e., full rank, the approximation error $\delta_{\mathrm{approx}} \to 0$, yielding the original upper bound in Theorem 1.

# 6 Experiments

We conduct extensive experiments to evaluate the proposed GADGET. We first introduce experiment setup in Section 6.1. Then, we provide the visualization of graph GDA to assess the necessity of incorporating GDA for graphs in Section 6.3. Afterwards, we evaluate GADGET's effectiveness on benchmark datasets in Section 6.2. We further conduct extensive studies on the varying shift levels (Section 6.4), hyperparameter sensitivity (Section 6.5), and path quality (Section 6.6).

## 6.1 Experimental Setup

We conduct extensive experiments on node classification using both synthetic and real-world datasets, including Airport Ribeiro et al. (2017), Citation Tang et al. (2008), Social Li et al. (2015), and contextual stochastic block model (CSBM) Deshpande et al. (2018). Airport dataset contains flight information of airports from Brazil, USA and Europe. Citation dataset includes academic networks from ACM and DBLP. Social dataset includes two blog networks from BlogCatalog (Blog1) Flickr (Blog2). We also adopt the CSBM model to generate various graph shifts, including attribute shifts with positively (Right) and negatively (Left) shifted attributes, degree shift with High and Low average degrees, and homophily shifts with high (Homo) and low (Hetero) homophilic scores in the source and target graphs. More details are in Appendix D.

We adopt two prominent GNN models, including GCN Kipf & Welling (2017) and APPNP Gasteiger et al. (2018), as the backbone classifier. Different adaptation baselines can be utilized to adapt knowledge from one graph to its consecutive graph along the path. Baseline adaptation methods include Empirical Risk Minimization (ERM), MMD Gretton et al. (2012), CORAL Sun et al. (2016), AdaGCN Dai et al. (2022), GRADE Wu et al. (2023) and StruRW Liu et al. (2023a).

During training, we have full access to source labels while having no knowledge on target labels. Results are averaged over five runs to avoid randomness. Our code is available at `https://github.com/zhichenz98/Gadget-TMLR`. More details are provided in Appendix D.

## 6.2 Effectiveness Results

To evaluate the effectiveness of GADGET in handling large shifts, we carry out experiment on both real-world and synthetic datasets, and the results are shown in Figure 2. In general, compared to direct adaptation (colored bars w/o hatches), we observe consistent improvements on the performance of a variety of graph DA methods and backbone GNNs on different datasets when applying GADGET (hatched bars). Specifically, on real-world datasets, GADGET achieves an average improvement of 6.77% on Airport, 3.58% on Social and 3.43% on Citation, compared to direct adaptation. On synthetic CSBM datasets, GADGET achieves more significant performance, improving various graph DA methods by 36.51% in average. More result statistics are provided in Appendix C.1.

Besides, we note a small number of cases where direct adaptation performs better than GADGET. According to Theorem 1, the target error depends on the source performance, the accumulated training error, and the generalization error. When the shift is mild, the accumulated training error dominates the error bound, so direct adaptation ($T = 1$) is preferable. For extremely large shifts, bridging the gap would require many steps, and the resulting linear growth of accumulated training error can outweigh the reduction of the generalization term. Therefore, it is essential to choose an appropriate $T$ that achieves a good balance between accumulated training error and generalization error.

## 6.3 Understanding the Gradual Adaptation Process

To better understand the necessity and mechanism of graph GDA, we first visualize the embedding spaces of the CSBM and Citation datasets trained under ERM. The results are shown in Figure 3, where different colors indicate different classes and different markers represent different domains.

Firstly, it is shown that large shifts exist in both datasets, as the source ($\bullet$) and target samples ($\times$) are scarcely overlapped. Besides that, direct adaptation often fails when facing large shifts. As shown in Figure 3, for the CSBM dataset, though the well-trained source model correctly classifies all source samples ($\textcolor{red}{\bullet},\textcolor{blue}{\bullet}$), all

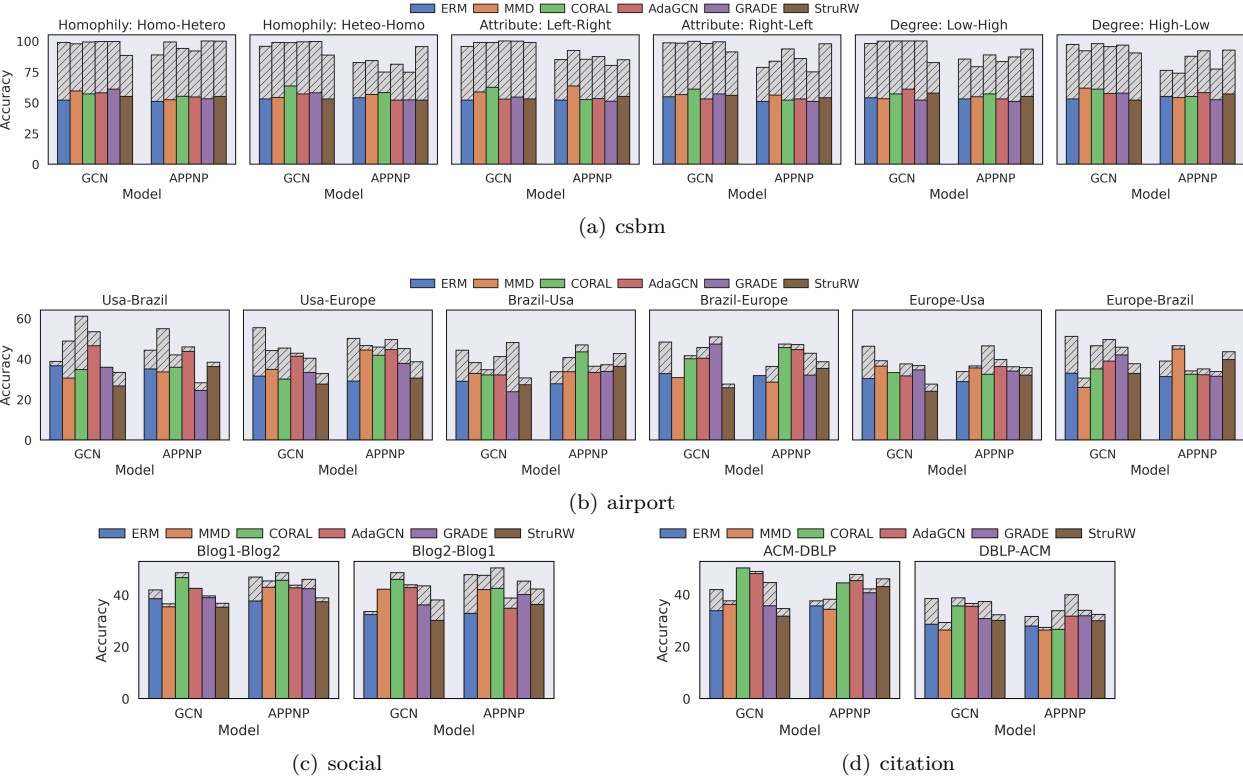

Figure 2: Experiment results. Different colors indicate different baseline adaptation methods. Bars with and without hatches indicate direct adaptation and gradual adaptation with GADGET, respectively. Our proposed GADGET consistently achieves better performance than direct adaptation on different backbone GNNs, adaptation methods and datasets. Best viewed in color.

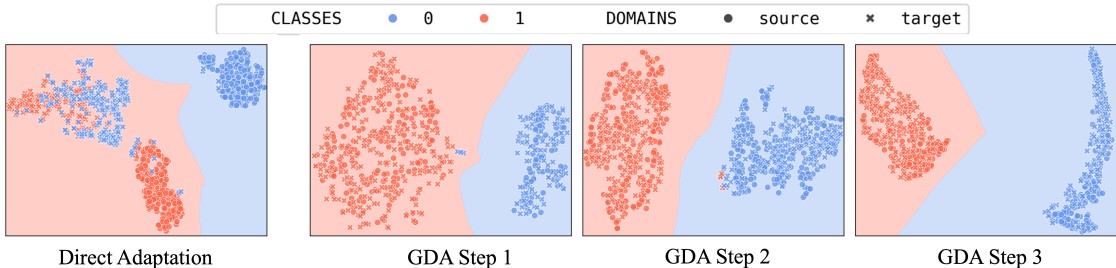

Figure 3: Embedding space of CSBM dataset under homophily shifts. Direct adaptation (left) fails when facing large shifts. GDA (right) correctly classifies most samples in each step, resulting in significant improvement in the classification accuracy. Best viewed in color.

target samples from class 0 ($\times$) are misclassified as class 1 ($\times$), due to the large shifts between the source and the target. In contrast, when adopting graph GDA, we expect smaller shifts between two successive domains, as source ($\bullet$) and target ($\times$) samples are largely overlapped, and the trained classifier correctly classifies most of the target samples.

The embedding space visualization provides further insights in the causes for performance degradation under large shifts, including representation degradation and classifier degradation. For representation degradation, we observe that although source samples are well separated, target samples are mixed together, indicating that source embedding transformation is suboptimal for the shifted target. For classifier degradation, while the classification boundary works well for source samples, it fails to classify target samples. However, when adopting GADGET, not only the target samples are well-separated, alleviating representation degradation, but also the classification boundary correctly classifies source and target samples, alleviating classifier degradation.

## 6.4 Mitigating Domain Shifts

To better understand how GADGET mitigates domain shifts, we test the GNN performance under various shift levels between source $\mathcal{G}_s$ and target $\mathcal{G}_t$. Specifically, we vary (1) the attribute shift level measured by $\Delta\boldsymbol{\mu} = |\mathrm{avg}(\boldsymbol{X}_s) - \mathrm{avg}(\boldsymbol{X}_t)|$, (2) the homophily shift level measured by $\Delta h = |h_s - h_t|$, and (3) the degree shift level measured by $\Delta d = |d_s - d_t|$. And the results are shown in Figure 4.

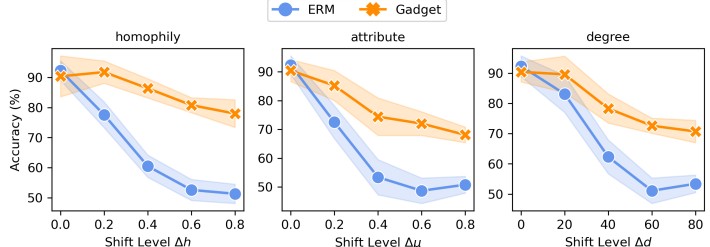

Figure 4: Classification accuracy under different shift levels.

As shown in the results, when the shift level increases, the performance of direct adaptation (ERM) drops rapidly, while the performance of gradual GDA (GADGET) is more robust. Compared to the performance under the mildest shift (left), ERM degrades up to 41.5% under the largest shift (right), behaving like random guessing as the classification accuracy approaches 0.5 on a binary classification task. However, GADGET only degrades up to 30.3% on the largest shift compared to performance under the mildest shift and outperforms ERM by up to 26.7% on the largest shift.

In addition, it is worth noting that GADGET underperforms direct adaptation when there domain shift does not exist. This is because the gradual GDA process involves self-training, which may introduce noisy pseudo-labels that mislead the training process. As we reveal in Theorem 2, the error bound includes an accumulated error $T\delta$. When domain shift is mild, i.e., $d_{\mathrm{FGW}}(\mathcal{G}_0, \mathcal{G}_1)$ is small, the effects of the accumulated error could be significant. And under such circumstances, as shown in Eq. equation 11, the optimal number of intermediate steps $T$ should be zero, i.e., direct adaptation.

## 6.5 Hyperparameter Study

We study how hyperparameters affect the performance and run time, including studies on the number $T$ of intermediate steps and the rank $r$ of low-rank OT. We experiment on the CSBM datasets with 500 nodes.

For the number of intermediate steps $T$, the results are shown in Figure 5(a). Overall, as $T$ increases, the performance first increases and then decreases, achieving the overall best performance when $T = 3$. This phenomenon aligns with our error bound in Theorem 2. When $T$ is smaller than the optimal $T$ in Eq. equation 11, the shifts between two successive graphs is large and the generalization error $\frac{C_{\mathrm{W}} \cdot d_{\mathrm{FGW}}^q(\mathcal{G}_0, \mathcal{G}_1)}{T^{q-1}}$ dominates the performance; Hence, the performance first improves. However, when $T$ is larger than the optimal $T$ in Eq. equation 11, the accumulated training error $T\delta$ dominates the performance; Hence, the performance degrades. Besides, we observe that the *training time* increases almost linearly w.r.t. $T$, as the gradual domain adaptation process involves repeated training the model for $T$ times. Based on the above observation, we choose $T = 3$ for the benchmark experiments as it achieves good trade-off between performance and efficiency.

For the choice of rank $r$, the results are in Figure 5(b). Overall, as $r$ increases, the performance first increases and then fluctuates at a high level. When $r$ is small, the transformation in Eq. equation 12 projects source and target graphs to small graphs, causing information loss during the transformation; Hence, the performance degrades. However, when $r$ is large enough, the transformation preserves most information in the source and target graphs; Hence achieving relatively stable performance. Besides, we observe that the *generation time* increases almost linear w.r.t. $r$, which aligns with our complexity analysis of $\mathcal{O}(Lndr + Ln^2r)$.

Besides, we study the effect of $\alpha$ on balancing the importance of node features and graph structure. We report the average performance on the Airport dataset in Figure 6. Overall, GADGET is relative robust to different selections with $\alpha \in (0, 1)$, under which the FGW distance consider both node features and graph structure. However, when $\alpha = 0$, i.e., Wasserstein distance considering features only, or $\alpha = 1$, i.e., GW distance considering structure only, we observe a performance degradation. This validates that both features and structure are crucial for the construction of the optimal path.

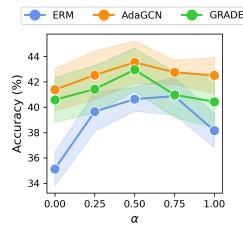

Figure 6: Study on $\alpha$.

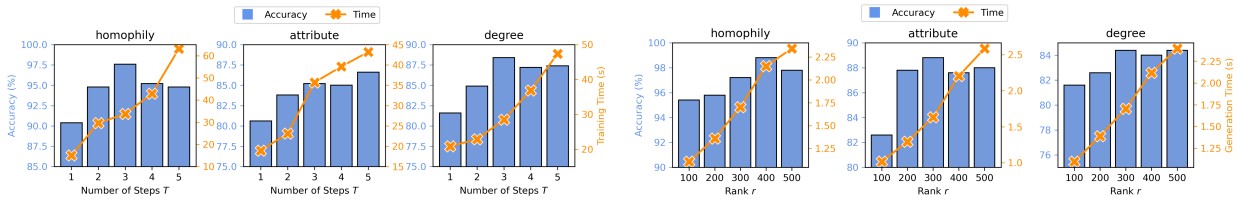

(a) Study on the number of intermediate steps $T$.

(b) Study on the rank $r$.

Figure 5: Hyperparameter Study.

## 6.6 Path quality Analysis

As Theorem 2 suggests, we expect the intermediate graphs lie on the FGW geodesics connecting source and target graphs. Following Definition 3, given any two values $\lambda_0, \lambda_1 \in [0, 1]$, the FGW distance between the generated graphs is expected to be proportional to the difference between the two values. We evaluate such correlation on the Citation dataset with results shown in Figure 7.

The results suggests that $d_{\mathrm{FGW}}(\gamma(\lambda_0), \gamma(\lambda_1))/d_{\mathrm{FGW}}(\mathcal{G}_0, \mathcal{G}_1)$ is strongly correlated with $|\lambda_0 - \lambda_1|$, with a Pearson correlation score of nearly 1. This validates that the generated graphs indeed lie along the FGW geodesics, thereby ensuring the effectiveness of graph GDA.

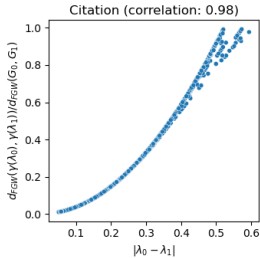

Figure 7: Path quality.

# 7 Conclusions

In this paper, we tackle large shifts on graphs, and propose GADGET, the first graph gradual domain adaptation framework to gradually adapt from source to target graph along the FGW geodesics. We establish a theoretical foundation by deriving an error bound for graph GDA based on the FGW discrepancy, motivated by which, we reveal that the optimal path minimizing the error bound lies on the FGW geodesics. A practical algorithm is further proposed to generate graphs on the FGW geodesics, complemented by entropy-based confidence for pseudo-label denoising, which enhances the self-training paradigm for graph GDA. Extensive experiments demonstrate the effectiveness of GADGET, enhancing various graph DA methods on different real-world datasets significantly.

# Acknowledgement

This work is supported by NSF (2118329, 2505932, 2537827, and 2416070) and AFOSR (FA9550-24-1-0002). The content of the information in this document does not necessarily reflect the position or the policy of the Government, and no official endorsement should be inferred. The U.S. Government is authorized to reproduce and distribute reprints for Government purposes notwithstanding any copyright notation here on.

# Broader Impact Statement

This paper presents work whose goal is to advance the field of Graph Machine Learning and Transfer Learning. There are many potential societal consequences of our work, none of which we feel must be specifically highlighted here.

# GenAI Usage Disclosure

In this manuscript, generative AI tool is used to edit and improve the quality of the text, including checking the spelling, grammar, punctuation and clarity.

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

# Appendix

## Contents

# A Proof

In this section, we provide detailed proof for all the Lemmas and Theorems. We first prove the Hölder continuity in Section A.1 (Lemma 1), then we prove the error bounds for node-level, edge-level and graph-level tasks in Section A.2 (Theorem 1). Finally, we prove the optimality of the FGW geodesic for graph GDA in Section A.3 (Theorems 2 and 3).

## A.1 Proof for Hölder Continuity

**Lemma 1** (Hölder continuity). *For any $L$-layer node-level GNN $f_n = \bigcirc_{l=L}^1 g^{(l)}$, edge-level GNN $f_e = \phi \circ \bigcirc_{l=L}^1 g^{(l)}$, and graph-level GNN $f_g = r \circ \bigcirc_{l=L}^1 g^{(l)}$, where $\bigcirc_{l=L}^1 g^{(l)} = g^{(L)} \circ \cdots \circ g^{(1)}$ and $g^{(i)}$ are GNN layers in equation 2. Given a source graph $\mathcal{G}_0$ and a target graph $\mathcal{G}_1$, we have:*

$$|\xi(f, \mathcal{G}_0)) - \xi(f, \mathcal{G}_1)| \le C \cdot d_{\mathrm{FGW};q,\beta}^q(\mathcal{G}_0, \mathcal{G}_1),$$

*where*

$$\beta = \frac{\alpha \left(1 - (C_c C_{\mathrm{lin}}(1-\alpha))^L\right)}{\alpha + (1-\alpha)^L (C_c C_{\mathrm{lin}})^{L-1}(1 - C_c C_{\mathrm{lin}})}$$

$$C_{\mathrm{gnn}} = C_c C_{\mathrm{lin}} \frac{\alpha + (1-\alpha)(C_c C_{\mathrm{lin}}(1-\alpha))^{L-1} - (C_c C_{\mathrm{lin}}(1-\alpha))^L}{1 - C_c C_{\mathrm{lin}}(1-\alpha)}$$

$$C = \begin{cases} C_{\mathrm{W}_n} C_{\mathrm{gnn}}, & \text{for node-level tasks} \\ 2C_{\mathrm{W}_e} C_\phi C_{\mathrm{gnn}}, & \text{for edge-level tasks} \\ C_{\mathrm{W}_g} C_r C_{\mathrm{gnn}}, & \text{for graph-level tasks} \end{cases}$$

*Proof.* We start with the node-level tasks based on Assumptions 1 and 2. Given two graphs $\mathcal{G}_0 = (\mathcal{V}_0, \boldsymbol{A}_0, \boldsymbol{X}_0), \mathcal{G}_1 = (\mathcal{V}_1, \boldsymbol{A}_1, \boldsymbol{X}_1)$. We denote the $l$-th layer embedding as $\boldsymbol{X}^{(l)} = f^{(l)} \circ \cdots \circ f^{(1)}(\mathcal{G})$, with corresponding graph $\mathcal{G}^{(l)} = (\mathcal{V}, \boldsymbol{A}, \boldsymbol{X}^{(l)})$. Let the marginal constraints be $\boldsymbol{\mu}_0 = \mathrm{Unif}(|\mathcal{V}_0|), \boldsymbol{\mu}_1 = \mathrm{Unif}(|\mathcal{V}_1|)$, for *any coupling* $\boldsymbol{\pi} \in \Pi(\boldsymbol{\mu}_0, \boldsymbol{\mu}_1)$, we have

$$\begin{aligned} &|\xi(f, \mathcal{G}_0) - \xi(f, \mathcal{G}_1)| \\ =& \left| \frac{1}{|\mathcal{V}_0|} \sum_{u \in \mathcal{V}_0} \epsilon(f, \{\boldsymbol{A}_0(u, u'), \boldsymbol{X}_0(u')\}_{u' \in \mathcal{V}_0}) - \frac{1}{|\mathcal{V}_1|} \sum_{v \in \mathcal{V}_1} \epsilon(f, \{\boldsymbol{A}_1(v, v'), \boldsymbol{X}_1(v')\}_{v' \in \mathcal{V}_1}) \right| \\ =& \left| \sum_{u \in \mathcal{V}_0} \boldsymbol{\mu}_0(u) \epsilon(f, \{\boldsymbol{A}_0(u, u'), \boldsymbol{X}_0(u')\}_{u' \in \mathcal{V}_0}) - \sum_{v \in \mathcal{V}_1} \boldsymbol{\mu}_1(v) \epsilon(f, \{\boldsymbol{A}_1(v, v'), \boldsymbol{X}_1(v')\}_{v' \in \mathcal{V}_1}) \right| \qquad (16) \\ =& \left| \mathbb{E}_{(u,v) \sim \boldsymbol{\pi}} \left( \epsilon(f, \{\boldsymbol{A}_0(u, u'), \boldsymbol{X}_0(u')\}_{u' \in \mathcal{V}_0}) - \epsilon(f, \{\boldsymbol{A}_1(v, v'), \boldsymbol{X}_1(v')\}_{v' \in \mathcal{V}_1}) \right) \right| \\ \le& \mathbb{E}_{(u,v) \sim \boldsymbol{\pi}} \left| \epsilon(f, \{\boldsymbol{A}_0(u, u'), \boldsymbol{X}_0(u')\}_{u' \in \mathcal{V}_0}) - \epsilon(f, \{\boldsymbol{A}_1(v, v'), \boldsymbol{X}_1(v')\}_{v' \in \mathcal{V}_1}) \right| \\ \le& \mathbb{E}_{(u,v) \sim \boldsymbol{\pi}} C_{\mathrm{W}_n} \|f(\mathcal{G}_0)_u - f(\mathcal{G}_1)_v\|_{\mathcal{Y}_n}^q \quad \text{(equation 6)} \end{aligned}$$

Now consider the $l$-th layer GNN $f^{(l)} = \mathrm{ReLU} \circ \mathrm{Linear} \circ g^{(l)}$, with input graph $\mathcal{G}^{(l-1)}$. For ReLU activation, given two inputs $\boldsymbol{X}_0, \boldsymbol{X}_1$, it is easy to show that

$$\|\mathrm{ReLU}(\boldsymbol{X}_0) - \mathrm{ReLU}(\boldsymbol{X}_1)\|_{\mathcal{X}} \le \|\boldsymbol{X}_0 - \boldsymbol{X}_1\|_{\mathcal{X}} \qquad (17)$$

For linear layer $\mathrm{Linear}(\boldsymbol{x}) = \boldsymbol{W}\boldsymbol{x} + \boldsymbol{b}$, given two inputs $\boldsymbol{X}_0, \boldsymbol{X}_1$, we can show that

$$\begin{aligned} \|\mathrm{Linear}(\boldsymbol{X}_0) - \mathrm{Linear}(\boldsymbol{X}_1)\|_{\mathcal{X}} &\le \|\boldsymbol{W}\| \|\boldsymbol{X}_0 - \boldsymbol{X}_1\|_{\mathcal{X}} \\ &\le C_{\mathrm{lin}} \|\boldsymbol{X}_0 - \boldsymbol{X}_1\|_{\mathcal{X}} \quad \text{(Assumption B)} \end{aligned} \qquad (18)$$

Combining equation 17 and equation 18, for any coupling $\boldsymbol{\pi} \in \Pi(\boldsymbol{\mu}_0, \boldsymbol{\mu}_1)$, we have

$$
\begin{aligned}
&\|f^{(l)}(\mathcal{G}_0^{(l-1)})_u - f^{(l)}(\mathcal{G}_1^{(l-1)})_v\|_{\mathcal{X}}^q \\
=&\|\text{ReLU} \circ \text{Linear} \circ g^{(l)}(\mathcal{G}^{l-1}) - \text{ReLU} \circ \text{Linear} \circ g^{(l)}(\mathcal{G}^{l-1})\|_{\mathcal{X}}^q \\
\leq&C_{\text{lin}}\|g^{(l)}(\mathcal{G}_0^{(l-1)})_u - g^{(l)}(\mathcal{G}_1^{(l-1)})_v\|_{\mathcal{X}}^q \\
\leq&C_{\text{c}}C_{\text{lin}}d_{\text{W}}^q\left(\mathcal{N}_{\mathcal{G}_0^{(l-1)}}(u), \mathcal{N}_{\mathcal{G}_1^{(l-1)}}(v)\right) \quad (\text{Assumption A}) \\
=&C_{\text{c}}C_{\text{lin}} \inf_{\boldsymbol{\tau} \in \Pi(\boldsymbol{\mu}_0, \boldsymbol{\mu}_1)} \mathbb{E}_{(u',v')\sim\boldsymbol{\tau}}\left[\alpha|\boldsymbol{A}_0(u,u') - \boldsymbol{A}_1(v,v')|^q + (1-\alpha)\|\boldsymbol{X}_0^{(l-1)}(u') - \boldsymbol{X}_1^{(l-1)}(v')\|_{\mathcal{X}}^q\right] \\
\leq&C_{\text{c}}C_{\text{lin}}\mathbb{E}_{(u',v')\sim\boldsymbol{\pi}}\left[\alpha|\boldsymbol{A}_0(u,u') - \boldsymbol{A}_1(v,v')|^q + (1-\alpha)\|\boldsymbol{X}_0^{(l-1)}(u') - \boldsymbol{X}_1^{(l-1)}(v')\|_{\mathcal{X}}^q\right] \\
=&C_{\text{c}}C_{\text{lin}}\left(\alpha\mathbb{E}_{(u',v')\sim\boldsymbol{\pi}}|\boldsymbol{A}_0(u,u') - \boldsymbol{A}_1(v,v')|^q + (1-\alpha)\|f^{(l-1)}(\mathcal{G}_0^{(l-2)})_u - f^{(l-1)}(\mathcal{G}_1^{(l-2)})_v\|_{\mathcal{X}}^q\right)
\end{aligned}
\tag{19}
$$

By repeatedly applying equation 19 to equation 16, we have:

$$
\begin{aligned}
&|\xi(f, \mathcal{G}_0) - \xi(f, \mathcal{G}_1)| \\
\leq&\ \mathbb{E}_{(u,v)\sim\boldsymbol{\pi}}C_{\text{W}_n}\|f(\mathcal{G}_0)_u - f(\mathcal{G}_1)_v\|_{\mathcal{Y}_n}^q \\
=&\ \mathbb{E}_{(u,v)\sim\boldsymbol{\pi}}C_{\text{W}_n}\|f^{(L)}(\mathcal{G}_0^{(L-1)})_u - f^{(L)}(\mathcal{G}_1)_v^{(L-1)}\|_{\mathcal{Y}_n}^q \\
\leq&\ C_{\text{W}_n}C_{\text{c}}C_{\text{lin}}\left(\alpha\mathbb{E}_{\substack{(u,v)\sim\boldsymbol{\pi}\\(u',v')\sim\boldsymbol{\pi}}}|\boldsymbol{A}_0(u,u') - \boldsymbol{A}_1(v,v')|^q + (1-\alpha)\mathbb{E}_{(u,v)\sim\boldsymbol{\pi}}\|f^{(L-1)}(\mathcal{G}_0^{(L-2)})_u - f^{(L-1)}(\mathcal{G}_1^{(L-2)})_v\|_{\mathcal{X}}^q\right) \quad (\text{equation 19}) \\
\leq&\ C_{\text{W}_n}C_{\text{c}}C_{\text{lin}}\left(\alpha\sum_{l=0}^{L-1}[C_{\text{c}}C_{\text{lin}}(1-\alpha)]^l\mathbb{E}_{\substack{(u,v)\sim\boldsymbol{\pi}\\(u',v')\sim\boldsymbol{\pi}}}|\boldsymbol{A}_0(u,u') - \boldsymbol{A}_1(v,v')|^q + (C_{\text{c}}C_{\text{lin}})^{L-1}(1-\alpha)^L\mathbb{E}_{(u,v)\sim\boldsymbol{\pi}}\|\boldsymbol{X}_0(u) - \boldsymbol{X}_1(v)\|_{\mathcal{X}}^q\right) \\
=&\ C_{\text{n}}\left(\beta\mathbb{E}_{\substack{(u,v)\sim\boldsymbol{\pi}\\(u',v')\sim\boldsymbol{\pi}}}|\boldsymbol{A}_0(u,u') - \boldsymbol{A}_1(v,v')|^q + (1-\beta)\mathbb{E}_{(u,v)\sim\boldsymbol{\pi}}\|\boldsymbol{X}_0(u) - \boldsymbol{X}_1(v)\|_{\mathcal{X}}^q\right) \\
\leq&\ C_{\text{n}} \inf_{\boldsymbol{\pi} \in \Pi(\boldsymbol{\mu}_0, \boldsymbol{\mu}_1)}\left(\beta\mathbb{E}_{\substack{(u,v)\sim\boldsymbol{\pi}\\(u',v')\sim\boldsymbol{\pi}}}|\boldsymbol{A}_0(u,u') - \boldsymbol{A}_1(v,v')|^q + (1-\beta)\mathbb{E}_{(u,v)\sim\boldsymbol{\pi}}\|\boldsymbol{X}_0(u) - \boldsymbol{X}_1(v)\|_{\mathcal{X}}^q\right) \\
=&\ C_{\text{n}}d_{\text{FGW};q,\beta}^q(\mathcal{G}_0^{(L)}, \mathcal{G}_1^{(L)})
\end{aligned}
\tag{20}
$$

where $\beta, C_{\text{n}}$ are defined as

$$
\begin{cases}
\beta = \dfrac{\alpha\left(1 - (C_{\text{c}}C_{\text{lin}}(1-\alpha))^L\right)}{\alpha + (1-\alpha)(C_{\text{c}}C_{\text{lin}}(1-\alpha))^{L-1} - (C_{\text{c}}C_{\text{lin}}(1-\alpha))^L} \\
C_{\text{n}} = C_{\text{W}_n}C_{\text{c}}C_{\text{lin}}\dfrac{\alpha + (1-\alpha)(C_{\text{c}}C_{\text{lin}}(1-\alpha))^{L-1} - (C_{\text{c}}C_{\text{lin}}(1-\alpha))^L}{1 - C_{\text{c}}C_{\text{lin}}(1-\alpha)}
\end{cases}
$$

To this point, we prove the node-level loss $\xi_n$ is Hölder continuous w.r.t. the $\beta$-FGW distance.

We can derive a similar proof for edge-level tasks. Let the marginal constraints be $\boldsymbol{\mu}_0 = \mathrm{Unif}(|\mathcal{V}_0|), \boldsymbol{\mu}_1 = \mathrm{Unif}(|\mathcal{V}_1|)$, for *any coupling* $\boldsymbol{\pi} \in \Pi(\boldsymbol{\mu}_0, \boldsymbol{\mu}_1)$, we have

$$
\begin{aligned}
&|\xi_e(f_e, \mathcal{G}_0) - \xi_e(f_e, \mathcal{G}_1)| \\
=& \left| \frac{1}{|\mathcal{V}_0|^2} \sum_{u,u' \in \mathcal{V}_0} \epsilon_e \left( f_e(\mathcal{G}_0)_{(u,u')} \right) - \frac{1}{|\mathcal{V}_1|^2} \sum_{v,v' \in \mathcal{V}_1} \epsilon_e \left( f_e(\mathcal{G}_1)_{(v,v')} \right) \right| \\
=& |\mathbb{E}_{u,u' \sim \boldsymbol{\mu}_0} \epsilon_e \left( f_e(\mathcal{G}_0)_{(u,u')} \right) - \mathbb{E}_{v,v' \sim \boldsymbol{\mu}_1} \epsilon_e \left( f_e(\mathcal{G}_1)_{(v,v')} \right)| \\
\leq& \mathbb{E}_{(u,v),(u',v') \sim \boldsymbol{\pi}} |\epsilon_e \left( f_e(\mathcal{G}_0)_{(u,u')} \right) - \epsilon_e \left( f_e(\mathcal{G}_1)_{(v,v')} \right)| \\
\leq& C_{\mathrm{W}_e} \cdot \mathbb{E}_{(u,v),(u',v') \sim \boldsymbol{\pi}} \| f_e(\mathcal{G}_0)_{(u,u')} - f_e(\mathcal{G}_1)_{(v,v')} \|_{\mathcal{Y}_e}^q \quad \text{(equation 7)} \\
=& C_{\mathrm{W}_e} \cdot \mathbb{E}_{(u,v),(u',v') \sim \boldsymbol{\pi}} \| \phi(f_n(\mathcal{G}_0)_u, f_n(\mathcal{G}_0)_{u'}) - \phi(f_n(\mathcal{G}_1)_v, f_n(\mathcal{G}_1)_{v'}) \|_{\mathcal{Y}_e}^q \\
\leq& C_{\mathrm{W}_e} C_\phi \cdot \mathbb{E}_{(u,v),(u',v') \sim \boldsymbol{\pi}} ( \| f_n(\mathcal{G}_0)_u - f_n(\mathcal{G}_1)_v \|_{\mathcal{X}}^q + \| f_n(\mathcal{G}_0)_{u'} - f_n(\mathcal{G}_1)_{v'} \|_{\mathcal{X}}^q ) \quad \text{(Assumption E)} \\
=& C_{\mathrm{W}_e} C_\phi \cdot \mathbb{E}_{(u,v),(u',v') \sim \boldsymbol{\pi}} ( \| g^{(L)}(\mathcal{G}_0^{(L-1)})_u - g^{(L)}(\mathcal{G}_1^{(L-1)})_v \|_{\mathcal{X}}^q + \| g^{(L)}(\mathcal{G}_0^{(L-1)})_{u'} - g^{(L)}(\mathcal{G}_1^{(L-1)})_{v'} \|_{\mathcal{X}}^q ) \\
=& 2 C_{\mathrm{W}_e} C_\phi \cdot \mathbb{E}_{(u,v) \sim \boldsymbol{\pi}} \| g^{(L)}(\mathcal{G}_0^{(L-1)})_u - g^{(L)}(\mathcal{G}_1^{(L-1)})_v \|_{\mathcal{X}}^q
\end{aligned}
$$

Similar to Lemma 1, we can leverage equation 19 and equation 20 to derive the following inequality

$$
\begin{aligned}
&|\xi_e(f_e, \mathcal{G}_0) - \xi_e(f_e, \mathcal{G}_1)| \\
=& 2 C_{\mathrm{W}_e} C_\phi \cdot \mathbb{E}_{(u,v) \sim \boldsymbol{\pi}} \| g^{(L)}(\mathcal{G}_0^{(L-1)})_u - g^{(L)}(\mathcal{G}_1^{(L-1)})_v \|_{\mathcal{X}}^q \\
\leq& 2 C_{\mathrm{W}_e} C_\phi C_c C_{\mathrm{lin}} \cdot d_{\mathrm{W}}^q \left( \mathcal{N}_{\mathcal{G}_0^{(L-1)}}(u), \mathcal{N}_{\mathcal{G}_1^{(L-1)}}(v) \right) \quad \text{(Assumption A)} \\
\leq& C_e \left( \beta \mathbb{E}_{\substack{(u,v) \sim \boldsymbol{\pi} \\ (u',v') \sim \boldsymbol{\pi}}} |\boldsymbol{A}_0(u,u') - \boldsymbol{A}_1(v,v')|^q + (1-\beta) \mathbb{E}_{(u,v) \sim \boldsymbol{\pi}} \| \boldsymbol{X}_0(u) - \boldsymbol{X}_1(v) \|_{\mathcal{X}}^q \right) \\
\leq& C_e \inf_{\boldsymbol{\pi} \in \Pi(\boldsymbol{\mu}_0, \boldsymbol{\mu}_1)} \left( \beta \mathbb{E}_{\substack{(u,v) \sim \boldsymbol{\pi} \\ (u',v') \sim \boldsymbol{\pi}}} |\boldsymbol{A}_0(u,u') - \boldsymbol{A}_1(v,v')|^q + (1-\beta) \mathbb{E}_{(u,v) \sim \boldsymbol{\pi}} \| \boldsymbol{X}_0(u) - \boldsymbol{X}_1(v) \|_{\mathcal{X}}^q \right) \\
=& C_e d_{\mathrm{FGW};q,}^q \left( \mathcal{G}_0^{(L)}, \mathcal{G}_1^{(L)} \right)
\end{aligned}
$$

where $\beta, C_e$ are defined as

$$
\begin{cases}
\beta = \dfrac{\alpha \left( 1 - (C_c C_{\mathrm{lin}}(1-\alpha))^L \right)}{\alpha + (1-\alpha)(C_c C_{\mathrm{lin}}(1-\alpha))^{L-1} - (C_c C_{\mathrm{lin}}(1-\alpha))^L} \\
C_e = 2 C_{\mathrm{W}_e} C_\phi C_c C_{\mathrm{lin}} \dfrac{\alpha + (1-\alpha)(C_c C_{\mathrm{lin}}(1-\alpha))^{L-1} - (C_c C_{\mathrm{lin}}(1-\alpha))^L}{1 - C_c C_{\mathrm{lin}}(1-\alpha)}
\end{cases}
$$

To this point, we prove the edge-level loss $\xi_e$ is Hölder continuous w.r.t. the $\beta$-FGW distance.

Finally, we prove for graph-level tasks

$$
|\xi_g(f_g, \mathcal{G}_0) - \xi_g(f_g, \mathcal{G}_1)| \leq C_{\mathrm{W}_g} \cdot \| f_g(\mathcal{G}_0) - f_g(\mathcal{G}_1) \|_{\mathcal{Y}}^q.
$$

Similar to the proof for Lemma 1, we can leverage equation 19 to derive the following inequality

$$
\begin{aligned}
|\xi_g(f_g, \mathcal{G}_0) - \xi_g(f_g, \mathcal{G}_1)| &\leq C_{\mathrm{W}_g} \cdot \|f_g(\mathcal{G}_0) - f_g(\mathcal{G}_1)\|_{\mathcal{Y}_g}^q \\
&= C_{\mathrm{W}_g} \|r \circ f(\mathcal{G}_0) - r \circ f(\mathcal{G}_1)\|_{\mathcal{X}} \\
&\leq C_{\mathrm{W}_g} C_{\mathrm{r}} \cdot \mathbb{E}_{(u,v)\sim\boldsymbol{\pi}} \|f(\mathcal{G}_0)_u - f(\mathcal{G}_1)_v\|_{\mathcal{X}} \quad \text{(Assumption F)} \\
&= C_{\mathrm{W}_g} C_{\mathrm{r}} \cdot \mathbb{E}_{(u,v)\sim\boldsymbol{\pi}} \|f^{(L)}(\mathcal{G}_0^{(L-1)})_u - f^{(L)}(\mathcal{G}_1^{(L-1)})_v\|_{\mathcal{X}} \\
&\leq C_{\mathrm{W}_g} C_{\mathrm{r}} C_{\mathrm{c}} C_{\mathrm{lin}} d_{\mathrm{W}}^q \left( \mathcal{N}_{\mathcal{G}_0^{(L-1)}}(u), \mathcal{N}_{\mathcal{G}_1^{(L-1)}}(v) \right) \quad \text{(Assumption A)} \\
&\leq C_{\mathrm{g}} \left( \beta \mathbb{E}_{\substack{(u,v)\sim\boldsymbol{\pi} \\ (u',v')\sim\boldsymbol{\pi}}} |\boldsymbol{A}_0(u,u') - \boldsymbol{A}_1(v,v')|^q + (1-\beta)\mathbb{E}_{(u,v)\sim\boldsymbol{\pi}} \|\boldsymbol{X}_0(u) - \boldsymbol{X}_1(v)\|_{\mathcal{X}}^q \right) \\
&\leq C_{\mathrm{g}} \inf_{\boldsymbol{\pi}\in\Pi(\boldsymbol{\mu}_0,\boldsymbol{\mu}_1)} \left( \beta \mathbb{E}_{\substack{(u,v)\sim\boldsymbol{\pi} \\ (u',v')\sim\boldsymbol{\pi}}} |\boldsymbol{A}_0(u,u') - \boldsymbol{A}_1(v,v')|^q + (1-\beta)\mathbb{E}_{(u,v)\sim\boldsymbol{\pi}} \|\boldsymbol{X}_0(u) - \boldsymbol{X}_1(v)\|_{\mathcal{X}}^q \right) \\
&= C_{\mathrm{g}} d_{\mathrm{FGW};q,\beta}^q (\mathcal{G}_0^{(L)}, \mathcal{G}_1^{(L)})
\end{aligned}
$$

where $\beta, C_{\mathrm{g}}$ are defined as

$$
\begin{cases}
\beta = \dfrac{\alpha \left(1 - (C_{\mathrm{c}}C_{\mathrm{lin}}(1-\alpha))^L\right)}{\alpha + (1-\alpha)(C_{\mathrm{c}}C_{\mathrm{lin}}(1-\alpha))^{L-1} - (C_{\mathrm{c}}C_{\mathrm{lin}}(1-\alpha))^L} \\[4mm]
C_{\mathrm{g}} = C_{\mathrm{W}_g} C_{\mathrm{r}} C_{\mathrm{c}} C_{\mathrm{lin}} \dfrac{\alpha + (1-\alpha)(C_{\mathrm{c}}C_{\mathrm{lin}}(1-\alpha))^{L-1} - (C_{\mathrm{c}}C_{\mathrm{lin}}(1-\alpha))^L}{1 - C_{\mathrm{c}}C_{\mathrm{lin}}(1-\alpha)}
\end{cases}
$$

To this point, we prove the graph-level loss $\xi_g$ is Hölder continuous w.r.t. the $\beta$-FGW distance. $\qquad\square$

## A.2  Proof for Error Bound

**Theorem 1** (Error bound). *Let $f_0$ denote the source model trained on the source graph $\mathcal{H}_0 = \mathcal{G}_0$. Suppose there are $T-1$ intermediate stages where in the $t$-th stage (for $t = 1, 2, ..., T$), we adapt $f_{t-1}$ to graph $\mathcal{H}_t$ to obtain an adapted $f_t$. If every adaptation step achieves $\|f_{t-1}(\mathcal{H}_t) - f_t(\mathcal{H}_t)\|_{\mathcal{Y}} \leq \delta$ on the corresponding graph $\mathcal{H}_t$, then the final error $\xi(f_T, \mathcal{H}_T)$ on target graph $\mathcal{H}_T = \mathcal{G}_1$ is upper bounded by*

$$
\xi(f_T, \mathcal{G}_1) \leq \xi(f_0, \mathcal{G}_0) + C_{\mathrm{f}} \cdot \delta T + C \cdot \sum_{t=1}^{T} d_{\mathrm{FGW};q,\beta}^q (\mathcal{H}_{t-1}, \mathcal{H}_t).
$$

*where $C_{\mathrm{f}} = C_{\mathrm{f}_n}$ for node-level task, $C_{\mathrm{f}} = C_{\mathrm{f}_e}$ for edge-level tasks, and $C_{\mathrm{f}} = C_{\mathrm{f}_g}$ for graph-level tasks.*

*Proof.* For any intermediate stage $t = 1, 2, ..., T$, we first consider node-level loss $\xi_n$:

$$
\begin{aligned}
&\left| \xi_n(f_{n_{t-1}}, \mathcal{H}_t) - \xi_n(f_{n_t}, \mathcal{H}_t) \right| \\
&= \left| \frac{1}{|\mathcal{V}_t|} \sum_{u\in\mathcal{V}_t} \epsilon_n(f_{n_{t-1}}, \mathcal{N}_{\mathcal{H}_t}(u)) - \frac{1}{|\mathcal{V}_t|} \sum_{u\in\mathcal{V}_t} \epsilon_n(f_{n_t}, \mathcal{N}_{\mathcal{H}_t}(u)) \right| \\
&\leq \frac{1}{|\mathcal{V}_t|} \sum_{u\in\mathcal{V}_t} \left| \epsilon_n(f_{n_{t-1}}, \mathcal{N}_{\mathcal{H}_t}(u)) - \epsilon_n(f_{n_t}, \mathcal{N}_{\mathcal{H}_t}(u)) \right| \\
&\leq \frac{1}{|\mathcal{V}_t|} \sum_{u\in\mathcal{V}_t} C_{\mathrm{f}_n} \cdot \left\| f_{n_{t-1}}(\mathcal{N}_{\mathcal{H}_t}(u)) - f_{n_t}(\mathcal{N}_{\mathcal{H}_t}(u)) \right\|_{\mathcal{Y}_n} \quad \text{(Assumption C)} \\
&= C_{\mathrm{f}_n} \cdot \frac{1}{|\mathcal{V}_t|} \sum_{u\in\mathcal{V}_t} \left\| f_{n_{t-1}}(\mathcal{H}_t)_u - f_{n_t}(\mathcal{H}_t)_u \right\|_{\mathcal{Y}_n} \\
&= C_{\mathrm{f}_n} \cdot \left\| f_{n_{t-1}}(\mathcal{H}_t) - f_{n_t}(\mathcal{H}_t) \right\|_{\mathcal{Y}_n} \\
&\leq C_{\mathrm{f}_n} \cdot \delta
\end{aligned}
\tag{21}
$$

Similarly, for edge-level loss $\xi_e$, we have:

$$
\begin{aligned}
&\left|\xi_e(f_{e_{t-1}}, \mathcal{H}_t) - \xi_e(f_{e_t}, \mathcal{H}_t)\right| \\
&= \left|\frac{1}{|\mathcal{V}_t|^2} \sum_{u,u' \in \mathcal{V}_t} \epsilon_e(f_{e_{t-1}}(\mathcal{H}_t)_{(u,u')}) - \frac{1}{|\mathcal{V}_t|^2} \sum_{u \in \mathcal{V}_t} \epsilon_e(f_{e_t}(\mathcal{H}_t)_{(u,u')})\right| \\
&\leq \frac{1}{|\mathcal{V}_t|^2} \sum_{u,u' \in \mathcal{V}_t} \left|\epsilon_e(f_{e_{t-1}}(\mathcal{H}_t)_{(u,u')}) - \sum_{u \in \mathcal{V}_t} \epsilon_e(f_{e_t}(\mathcal{H}_t)_{(u,u')})\right| \\
&\leq \frac{1}{|\mathcal{V}_t|^2} \sum_{u \in \mathcal{V}_t} C_{\mathrm{f}_e} \cdot \left\|f_{e_{t-1}}(\mathcal{H}_t)_{(u,u')} - f_{e_t}(\mathcal{H}_t)_{(u,u')}\right\|_{\mathcal{Y}_e} \quad \text{(Assumption C)} \\
&= C_{\mathrm{f}_e} \cdot \frac{1}{|\mathcal{V}_t|^2} \sum_{u \in \mathcal{V}_t} \left\|f_{e_{t-1}}(\mathcal{H}_t)_{(u,u')} - f_{e_t}(\mathcal{H}_t)_{(u,u')}\right\|_{\mathcal{Y}_e} \\
&= C_{\mathrm{f}_e} \cdot \left\|f_{e_{t-1}}(\mathcal{H}_t) - f_{e_t}(\mathcal{H}_t)\right\|_{\mathcal{Y}_e} \\
&\leq C_{\mathrm{f}_e} \cdot \delta
\end{aligned}
\tag{22}
$$

Similarly, for graph-level loss $\xi_g$, we have:

$$
\begin{aligned}
&\left|\xi_g(f_{g_{t-1}}, \mathcal{H}_t) - \xi_g(f_{e_t}, \mathcal{H}_t)\right| \\
&\leq C_{\mathrm{f}_g} \cdot \left\|f_{e_{t-1}}(\mathcal{H}_t) - f_{e_t}(\mathcal{H}_t)\right\|_{\mathcal{Y}_g} \quad \text{(Assumption C)} \\
&\leq C_{\mathrm{f}_g} \cdot \delta
\end{aligned}
\tag{23}
$$

For simplicity, we slightly abuse $C_{\mathrm{f}}$ as a general notation for $C_{\mathrm{f}_n}, C_{\mathrm{f}_e}, C_{\mathrm{f}_g}$, and abuse $f$ as a general notation for $f_n, f_e, f_g$. Therefore, equation 21, equation 22, and equation 23 can be uniformly written as

$$
|\xi(f_{t-1}, \mathcal{H}_t) - \xi(f_t, \mathcal{H}_t)| \leq C_{\mathrm{f}} \cdot \delta
\tag{24}
$$

Based on equation 24 and Lemma 1, we have

$$
\begin{aligned}
&|\xi(f_{t-1}, \mathcal{H}_{t-1}) - \xi(f_t, \mathcal{H}_t)| \\
&\leq |\xi(f_{t-1}, \mathcal{H}_{t-1}) - \xi(f_t, \mathcal{H}_{t-1})| + |\xi(f_t, \mathcal{H}_{t-1}) - \xi(f_t, \mathcal{H}_t)| \\
&\leq C_{\mathrm{f}} \cdot \delta + C \cdot d^q_{\mathrm{FGW};q,\beta}(\mathcal{H}_{t-1}, \mathcal{H}_t)
\end{aligned}
$$

Therefore, we have:

$$
\begin{aligned}
\xi(f_T, \mathcal{G}_1) &= \xi(f_T, \mathcal{H}_T) \\
&= \xi(f_0, \mathcal{H}_0) + |\xi(f_T, \mathcal{H}_T) - \xi(f_0, H_0)| \\
&= \xi(f_0, \mathcal{H}_0) + \left|\sum_{t=1}^{T} (\xi(f_{t-1}, \mathcal{H}_t) - \xi(f_t, \mathcal{H}_t))\right| \\
&\leq \xi(f_0, \mathcal{H}_0) + \sum_{t=1}^{T} |\xi(f_{t-1}, \mathcal{H}_t) - \xi(f_t, \mathcal{H}_t)| \\
&\leq \xi(f_0, \mathcal{H}_0) + \sum_{t=1}^{T} \left(C_{\mathrm{f}} \cdot \delta + C \cdot d^q_{\mathrm{FGW};q,\beta}(\mathcal{H}_{t-1}, \mathcal{H}_t)\right) \\
&= \xi(f_0, \mathcal{H}_0) + C_{\mathrm{f}} \cdot \delta T + C \sum_{t=1}^{T} d^q_{\mathrm{FGW};q,\beta}(\mathcal{H}_{t-1}, \mathcal{H}_t) \\
&= \xi(f_0, \mathcal{G}_0) + C_{\mathrm{f}} \cdot \delta T + C \sum_{t=1}^{T} d^q_{\mathrm{FGW};q,\beta}(\mathcal{H}_{t-1}, \mathcal{H}_t)
\end{aligned}
$$

$\square$

## A.3 Proof for Optimal Path

**Theorem 2** (Optimal path). *Given a source graph $\mathcal{G}_0$ and a target graph $\mathcal{G}_1$, let $\gamma : [0,1] \to \mathfrak{G}/\sim$ be an FGW geodesic connecting $\mathcal{G}_0$ and $\mathcal{G}_1$. Then the error bound in Theorem 1 attains its **minimum** when intermediate graphs are $\mathcal{H}_t = \gamma(\frac{t}{T}), \forall t = 0, 1, ..., T$, where we have:*

$$\xi(f_T, \mathcal{G}_1) \le \xi(f_0, \mathcal{G}_0) + C_{\mathrm{f}} \cdot \delta T + \frac{C \cdot d^q_{\mathrm{FGW};q,\beta}(\mathcal{G}_0, \mathcal{G}_1)}{T^{q-1}}.$$

*Proof.* Note that for any intermediate graphs $\mathcal{H}_1, ..., \mathcal{H}_{T-1}$, by Jensen's inequality of the convex function $z \to |z|^q$ and the triangle inequality of $d_{\mathrm{FGW}}$, we have:

$$\begin{aligned}
\sum_{t=1}^{T} d^q_{\mathrm{FGW};q,\beta}(\mathcal{H}_{t-1}, \mathcal{H}_t) &= T \sum_{t=1}^{T} \frac{d^q_{\mathrm{FGW};q,\beta}(\mathcal{H}_{t-1}, \mathcal{H}_t)}{T} \\
&\ge T \sum_{t=1}^{T} \left( \frac{d_{\mathrm{FGW};q,\beta}(\mathcal{H}_{t-1}, \mathcal{H}_t)}{T} \right)^q \\
&= \frac{\sum_{t=1}^{T} d^q_{\mathrm{FGW};q,\beta}(\mathcal{H}_{t-1}, \mathcal{H}_t)}{T^{q-1}} \\
&\ge \frac{d^q_{\mathrm{FGW};q,\beta}(\mathcal{H}_{t-1}, \mathcal{H}_t)}{T^{q-1}}
\end{aligned}$$

When the intermediate graphs $\mathcal{H}_t, \forall t = 1, 2, ..., T$ are on the FGW geodesics, i.e., $\mathcal{H}_t = \gamma\left(\frac{t}{T}\right)$, by the geodesic property in Definition 3, we have

$$\begin{aligned}
d_{\mathrm{FGW};q,\beta}(\mathcal{H}_{t-1}, \mathcal{H}_t) &= d_{\mathrm{FGW};q,\beta}\left( \gamma\left(\frac{t-1}{T}\right), \gamma\left(\frac{t}{T}\right) \right) \\
&= \left| \frac{t-1}{T} - \frac{t}{T} \right| \cdot d_{\mathrm{FGW};q,\beta}(\gamma(0), \gamma(1)) \\
&= \frac{1}{T} \cdot d_{\mathrm{FGW};q,\beta}(\mathcal{G}_0, \mathcal{G}_1)
\end{aligned}$$

Therefore, we have

$$\begin{aligned}
\xi(f_T, \mathcal{G}_1) &\le \xi(f_0, \mathcal{G}_0) + C_{\mathrm{f}_n} \cdot \delta T + C \sum_{t=1}^{T} d^q_{\mathrm{FGW};q,\beta}(\mathcal{H}_{t-1}, \mathcal{H}_t) \\
&= \xi(f_0, \mathcal{G}_0) + C_{\mathrm{f}_n} \cdot \delta T + C \sum_{t=1}^{T} \left( \frac{1}{T} d_{\mathrm{FGW};q,\beta}(\mathcal{G}_0, \mathcal{G}_1) \right)^q \\
&= \xi(f_0, \mathcal{G}_0) + C_{\mathrm{f}_n} \cdot \delta T + \frac{C \cdot d^q_{\mathrm{FGW};q,\beta}(\mathcal{G}_0, \mathcal{G}_1)}{T^{q-1}}
\end{aligned}$$

which realize the lower bound. Therefore, the geodesic $\gamma$ gives the optimal path for graph GDA. $\square$

**Theorem 3** (FGW geodesic). *Given a source graph $\mathcal{G}_0$ and a target graph $\mathcal{G}_1$, the transformed graphs $\tilde{\mathcal{G}}_0, \tilde{\mathcal{G}}_1$ are in the FGW equivalent class of $\mathcal{G}_0, \mathcal{G}_1$, i.e., $[\![\mathcal{G}_0]\!] = [\![\tilde{\mathcal{G}}_0]\!], [\![\mathcal{G}_1]\!] = [\![\tilde{\mathcal{G}}_1]\!]$. Besides that, the intermediate graphs $\mathcal{H}_t, \forall t = 0, 1, ..., T$, generated by equation 13 are on an FGW geodesic connecting $\mathcal{G}_0$ and $\mathcal{G}_1$.*

*Proof.* Given a source graph $\mathcal{G}_0 = (\mathcal{V}_0, \boldsymbol{A}_0, \boldsymbol{X}_0)$ and a target graph $\mathcal{G}_1 = (\mathcal{V}_1, \boldsymbol{A}_1, \boldsymbol{X}_1)$, as well as their probability measures $\boldsymbol{\mu}_1, \boldsymbol{\mu}_2$, we obtain the optimal FGW matching $\boldsymbol{S}$ based on equation 1.

We first show that the transformed graphs $\tilde{\mathcal{G}}_0, \tilde{\mathcal{G}}_1$ from equation 12 are in the FGW equivalent classes of $\mathcal{G}_0, \mathcal{G}_1$, respectively. The transformed graphs are on the product space of $\mathcal{G}_0$ and $\mathcal{G}_1$, and we can write out

the FGW distance between $\mathcal{G}_0$ and $\tilde{\mathcal{G}}_0$ as follows

$$d_{\mathrm{FGW}}(\mathcal{G}_0, \tilde{\mathcal{G}}_0)$$
$$= \min_{\boldsymbol{S} \in \Pi(\boldsymbol{\mu}_1, \tilde{\boldsymbol{\mu}}_1)} (1-\alpha)\mathbb{E}_{(u,(x,y)) \sim \boldsymbol{S}} \boldsymbol{M}(u,(x,y))^q + \alpha \mathbb{E}_{\substack{(u,(x,y)) \sim \boldsymbol{S} \\ (u',(x',y')) \sim \boldsymbol{S}}} |\boldsymbol{A}_0(u,u') - \tilde{\boldsymbol{A}}_0((x,y),(x',y'))|^q$$

Consider a the following naive coupling satisfying the marginal constraint $\boldsymbol{S} \in \Pi(\boldsymbol{\mu}_0, \tilde{\boldsymbol{\mu}}_0)$

$$\boldsymbol{S}(u,(x,y)) = \begin{cases} \dfrac{\boldsymbol{\mu}_0(u)}{|\mathcal{V}_1|}, & \text{if } u = x \\ 0, & \text{else} \end{cases}, \tag{25}$$

the FGW distance $d_{\mathrm{FGW}}(\mathcal{G}_0, \tilde{\mathcal{G}}_0)$ with optimal coupling $\boldsymbol{S}^*$ is upper bounded by $\varepsilon_{\mathcal{G}_0, \tilde{\mathcal{G}}_0}(\boldsymbol{S}_0)$ as follows

$$d_{\mathrm{FGW}}^q(\mathcal{G}_0, \tilde{\mathcal{G}}_0)$$
$$= (1-\alpha)\mathbb{E}_{(u,(x,y)) \sim \boldsymbol{S}^*} \boldsymbol{M}(u,(x,y)) + \alpha \mathbb{E}_{\substack{(u,(x,y)) \sim \boldsymbol{S}^* \\ (u',(x',y')) \sim \boldsymbol{S}^*}} |\boldsymbol{A}_0(u,u') - \tilde{\boldsymbol{A}}_0((x,y),(x',y'))|^q$$

$$= (1-\alpha)\mathbb{E}_{(u,(x,y)) \sim \boldsymbol{S}^*} \left| \boldsymbol{X}(u) - \sum_{i \in \mathcal{V}_0} \boldsymbol{P}_0(i,(x,y))\boldsymbol{X}(i) \right|^q + \alpha \mathbb{E}_{\substack{(u,(x,y)) \sim \boldsymbol{S}^* \\ (u',(x',y')) \sim \boldsymbol{S}^*}} \left| \boldsymbol{A}_0(u,u') - \sum_{\substack{i \in \mathcal{V}_0 \\ j \in \mathcal{V}_1}} \boldsymbol{P}_0(i,(x,y))\boldsymbol{A}_0(i,j)\boldsymbol{P}_0(j,(x',y')) \right|^q$$

$$= (1-\alpha)\mathbb{E}_{(u,(x,y)) \sim \boldsymbol{S}^*} |\boldsymbol{X}(u) - \boldsymbol{X}(x)|^q + \alpha \mathbb{E}_{\substack{(u,(x,y)) \sim \boldsymbol{S}^* \\ (u',(x',y')) \sim \boldsymbol{S}^*}} |\boldsymbol{A}_0(u,u') - \boldsymbol{A}_0(x,x')|^q \quad \text{(equation 12)}$$

$$\leq (1-\alpha)\mathbb{E}_{(u,(x,y)) \sim \boldsymbol{S}_0} |\boldsymbol{X}(u) - \boldsymbol{X}(u)|^q + \alpha \mathbb{E}_{\substack{(u,(x,y)) \sim \boldsymbol{S}_0 \\ (u',(x',y')) \sim \boldsymbol{S}_0}} |\boldsymbol{A}_0(u,u') - \boldsymbol{A}_0(u,u')|^q \quad \text{(equation 25)}$$

$$= 0$$

Due to the non-negativity property of the FGW distance Vayer et al. (2020), we prove that $d_{\mathrm{FGW}}(\mathcal{G}_0, \tilde{\mathcal{G}}_0) = 0$, i.e., $\mathcal{G}_0 \sim \tilde{\mathcal{G}}_0$. Similarly, we can show that $\mathcal{G}_1 \sim \tilde{\mathcal{G}}_1$.

Afterwards, we prove that the interpolation in equation 12 and equation 13 generate intermediate graphs on the FGW geodesics. According to Vayer et al. (2020), the FGW geodesics connecting $\mathcal{G}_0$ and $\mathcal{G}_1$ is a graph in the product space $\mathcal{G} = (\mathcal{V}_0 \otimes \mathcal{V}_1, \tilde{\boldsymbol{A}}, \tilde{\boldsymbol{X}})$ satisfying the following property:

$$\tilde{\mathcal{G}}_{\frac{t}{T}} = (\tilde{\mathcal{V}}_{\frac{t}{T}}, \tilde{\boldsymbol{A}}_{\frac{t}{T}}, \tilde{\boldsymbol{X}}_{\frac{t}{T}})$$

$$\text{where} \quad \begin{cases} \tilde{\mathcal{V}}_{\frac{t}{T}} = \mathcal{V}_0 \otimes \mathcal{V}_1 \\ \tilde{\boldsymbol{A}}_{\frac{t}{T}}((u,v),(u',v')) = \left(1 - \dfrac{t}{T}\right)\boldsymbol{A}_0(u,u') + \dfrac{t}{T}\boldsymbol{A}_1(v,v'), \forall u, u' \in \mathcal{V}_0, v, v' \in \mathcal{V}_1 \\ \tilde{\boldsymbol{X}}_{\frac{t}{T}}((u,v)) = \left(1 - \dfrac{t}{T}\right)\boldsymbol{X}_0(u) + \dfrac{t}{T}\boldsymbol{X}_1(v), \forall u \in \mathcal{V}_0, v \in \mathcal{V}_1 \end{cases} \tag{26}$$

Following the transformation in equation 12, for nodes $u, u' \in \mathcal{V}_0$ and $v, v' \in \mathcal{V}_1$, we can rewrite the transformed adjacency matrix $\tilde{\boldsymbol{A}}_0$ and attribute matrix $\tilde{\boldsymbol{X}}_0$ as follows

$$\tilde{\boldsymbol{A}}_0((u,v),(u',v')) = \sum_{i \in \mathcal{V}_0, j \in \mathcal{V}_1} \boldsymbol{P}_0(i,(u,v))\boldsymbol{A}_0(i,j)\boldsymbol{P}_1(j,(u',v')) = \boldsymbol{A}_0(u,u')$$

$$\tilde{\boldsymbol{X}}_0((u,v)) = \sum_{i \in \mathcal{V}_0} \boldsymbol{P}_0(i,(u,v))\boldsymbol{X}_0(i) = \boldsymbol{X}_0(u)$$

Therefore, the intermediate graph $\mathcal{H}_t$ in equation 13 can be expresserd by:

$$\mathcal{H}_t = (\mathcal{V}_{\frac{t}{T}}, \tilde{\boldsymbol{A}}_{\frac{t}{T}}, \tilde{\boldsymbol{X}}_{\frac{t}{T}})$$

$$\text{where} \quad \begin{cases} \mathcal{V}_{\frac{t}{T}} = \mathcal{V}_0 \otimes \mathcal{V}_1 \\ \tilde{\boldsymbol{A}}_{\frac{t}{T}}((u,v),(u',v')) = \left(1 - \dfrac{t}{T}\right)\boldsymbol{A}_0(u,u') + \dfrac{t}{T}\boldsymbol{A}_1(v,v') \\ \tilde{\boldsymbol{X}}_{\frac{t}{T}}((u,v)) = \left(1 - \dfrac{t}{T}\right)\boldsymbol{X}_0(u) + \dfrac{t}{T}\boldsymbol{X}_1(v) \end{cases}$$

Now, we consider a naive "diagonal" coupling $\pi_{t_1,t_2}$ between $\mathcal{H}_{t_1}, \mathcal{H}_{t_2}$ as follows

$$\pi_{t_1,t_2}((u,v),(u',v')) = \begin{cases} \pi(u,v), & \text{if } u = u' \text{ and } v = v' \\ 0, & \text{else} \end{cases}$$

Afterwards, the FGW distance between $\mathcal{H}_{t_1}$ and $\mathcal{H}_{t_2}$ should be less or equal to the FGW distance under the 'diagonal' coupling, that is:

$$\begin{aligned} & d_{\text{FGW}}(\mathcal{H}_{t_1}, \mathcal{H}_{t_2}) \\ & \leq \sum_{u,v,u',v'} \left[ (1-\alpha)|\tilde{\boldsymbol{X}}_{\frac{t_1}{T}}(u) - \tilde{\boldsymbol{X}}_{\frac{t_2}{T}}(u')| + \alpha \cdot |\tilde{\boldsymbol{A}}_{\frac{t_1}{T}}(u,v) - \tilde{\boldsymbol{A}}_{\frac{t_2}{T}}(u',v')| \right] \pi_{t_1,t_2}((u,v),(u',v')) \\ & = \sum_{u,v} \left[ (1-\alpha)|\tilde{\boldsymbol{X}}_{\frac{t_1}{T}}(u) - \tilde{\boldsymbol{X}}_{\frac{t_2}{T}}(u)| + \alpha \cdot |\tilde{\boldsymbol{A}}_{\frac{t_1}{T}}(u,v) - \tilde{\boldsymbol{A}}_{\frac{t_2}{T}}(u,v)| \right] \pi(u,v) \end{aligned}$$

According to equation 26, we have

$$\left| \tilde{\boldsymbol{X}}_{\frac{t_1}{T}}(u) - \tilde{\boldsymbol{X}}_{\frac{t_2}{T}}(u) \right| = \left| (1 - \frac{t_1}{T})\boldsymbol{X}_0(u) + \frac{t_1}{T}\boldsymbol{X}_1(u) - (1 - \frac{t_2}{T})\boldsymbol{X}_2(u) - \frac{t_2}{T}\boldsymbol{X}_2(u) \right| = \left| \frac{t_1 - t_2}{T} \right| \cdot |\boldsymbol{X}_0(u) - \boldsymbol{X}_1(u)|$$

$$\left| \boldsymbol{A}_{\frac{t_1}{T}}(u,v) - \boldsymbol{A}_{\frac{t_2}{T}}(u,v) \right| = \left| (1 - \frac{t_1}{T})\boldsymbol{A}_0(u,v) + \frac{t_1}{T}\boldsymbol{A}_1(u,v) - (1 - \frac{t_2}{T})\boldsymbol{A}_2(u,v) - \frac{t_2}{T}\boldsymbol{A}_2(u,v) \right| = \left| \frac{t_1 - t_2}{T} \right| \cdot |\boldsymbol{A}_0(u,v) - \boldsymbol{A}_1(u,v)|$$

Combine the above two equations, we have

$$\begin{aligned} & d_{\text{FGW}}(\mathcal{H}_{t_1}, \mathcal{H}_{t_2}) \\ & \leq \left| \frac{t_1 - t_2}{T} \right| \cdot \sum_{u,v} [(1-\alpha)|\boldsymbol{X}_0(u) - \boldsymbol{X}_1(u)| + \alpha \cdot |\boldsymbol{A}_0(u,v) - \boldsymbol{A}_1(u,v)|]\pi(u,v) \\ & = \left| \frac{t_1 - t_2}{T} \right| d_{\text{FGW}}(G_0, G_1) \end{aligned} \tag{27}$$

The above inequality holds for any $0 \leq \frac{t_1}{T} \leq \frac{t_2}{T} \leq 1$. In particular, we have

$$d_{\text{FGW}}(\mathcal{G}_0, \mathcal{H}_{t_1}) \leq \left| 0 - \frac{t_1}{T} \right| d_{\text{FGW}}(\mathcal{G}_0, \mathcal{G}_1) = \frac{t_1}{T} d_{\text{FGW}}(\mathcal{G}_0, \mathcal{G}_1)$$

$$d_{\text{FGW}}(\mathcal{H}_{t_1}, \mathcal{H}_{t_2}) \leq \left| \frac{t_1}{T} - \frac{t_2}{T} \right| d_{\text{FGW}}(\mathcal{G}_0, \mathcal{G}_1) = \frac{t_2 - t_1}{T} d_{\text{FGW}}(\mathcal{G}_0, \mathcal{G}_1)$$

$$d_{\text{FGW}}(\mathcal{H}_{t_2}, \mathcal{G}_1) \leq \left| \frac{t_2}{T} - 1 \right| d_{\text{FGW}}(\mathcal{G}_0, \mathcal{G}_1) = (1 - \frac{t_2}{T}) d_{\text{FGW}}(\mathcal{G}_0, \mathcal{G}_1)$$

Finally, by the triangle inequality of FGW distance Vayer et al. (2020), we have

$$d_{\text{FGW}}(\mathcal{G}_0, \mathcal{G}_1) \leq \frac{t_1}{T} d_{\text{FGW}}(\mathcal{G}_0, \mathcal{G}_1) + \frac{t_2 - t_1}{T} d_{\text{FGW}}(\mathcal{G}_0, \mathcal{G}_1) + (1 - \frac{t_2}{T}) d_{\text{FGW}}(\mathcal{G}_0, \mathcal{G}_1) = d_{\text{FGW}}(\mathcal{G}_0, \mathcal{G}_1)$$

Hence, the $\leq$ in this inequality is actually =; in particular, $d_{\text{FGW}}(\mathcal{H}_{t_1}, \mathcal{H}_{t_2}) = |\frac{t_1}{T} - \frac{t_2}{T}| d_{\text{FGW}}(\mathcal{G}_0, \mathcal{G}_1)$. Therefore, we prove that the intermediate graphs $\mathcal{H}_t$ generated by equation 13 are on the FGW geodesics connecting $\mathcal{G}_0$ and $\mathcal{G}_1$. $\square$

### A.4 Proof for practical error bound

**Theorem 4** (Practical error bound). *For a sequence of intermediate graphs $\tilde{H}_0, \ldots, \tilde{H}_T$ on the approximated FGW geodesics generated by* GADGET, *performing GDA along the path yield a final error $\xi(f_T, \mathcal{H}_T)$ on target graph $\mathcal{H}_T = \mathcal{G}_1$ upper bounded by*

$$\xi(f_T, \mathcal{G}_1) \leq \text{Original bound} + 4C\delta_{approx} \sum_{t=1}^{T} d_{FGW}(\mathcal{H}_t, \mathcal{H}_{t+1}) + 4CT\delta_{approx}^2$$

*where original bound is the upper bound on the exact geodesics provided in Theorem 1, and $\delta_{approx} = \max_t d_{FGW}(\mathcal{H}_t, \tilde{H}_t)$ is the maximum approximation error between exact geodesic graph $\mathcal{H}_t$ and low-rank approximated graph $\tilde{\mathcal{H}}_t$.*

*Proof.* Let $\mathcal{H}$ be the graph on the exact geodesic and $\tilde{\mathcal{H}}$ be the graph on the approximate path with rank-$r$. We use $\boldsymbol{P}$ to denote the optimal coupling and $\tilde{\boldsymbol{P}}$ to denote the approximated low-rank coupling. We denote $\Delta_{\boldsymbol{P}} = \|\boldsymbol{P} - \tilde{\boldsymbol{P}}\|_F$. We denote $\delta_{\text{approx}} = \max_t d_{\text{FGW}}(\mathcal{H}_t, \tilde{\mathcal{H}}_t)$ as the maximum approximation error between the exact geodesic graph $\mathcal{H}_t$ (full-rank) and approximated geodesic graph $\tilde{\mathcal{H}}_t$.

First, we bound the approximation gap $\delta_{\text{approx}}$. When considering a naive identity coupling (i.e., the $i$-th node in $\mathcal{H}_t$ align with the $i$-th node in $\tilde{\mathcal{H}}_t$) between $\mathcal{H}_t$ and $\tilde{\mathcal{H}}_t$, it induces an upper bound of the FGW distance as

$$d_{\text{FGW}}^2(\mathcal{H}_t, \tilde{\mathcal{H}}_t) \leq (1-\alpha)\|\boldsymbol{X}_t - \tilde{\boldsymbol{X}}_t\|_F + \alpha\|\boldsymbol{A}_t - \tilde{\boldsymbol{A}}_t\|_F$$

According to the transformation in Eq. 13, feature error at step $t$ can be written as

$$\|\boldsymbol{X}_t - \tilde{\boldsymbol{X}}_t\|_F = \|(1-\frac{t}{T})(\boldsymbol{P}_0^\mathsf{T} - \tilde{\boldsymbol{P}}_0^\mathsf{T})\boldsymbol{X}_0 + \frac{t}{T}(\boldsymbol{P}_1^\mathsf{T} - \tilde{\boldsymbol{P}}_1^\mathsf{T})\boldsymbol{X}_1\|_F \leq (1-\frac{t}{T})\Delta_{\boldsymbol{P}_0}\|\boldsymbol{X}_0\|_F + \frac{t}{T}\Delta_{\boldsymbol{P}_1}\|\boldsymbol{X}_1\|_F$$

Similarly, the structure error at step $t$ can be written as

$$\|\boldsymbol{A}_t - \tilde{\boldsymbol{A}}_t\|_F = \|(1-\frac{t}{T})(\boldsymbol{P}_0^\mathsf{T}\boldsymbol{A}_0\boldsymbol{P}_0 - \tilde{\boldsymbol{P}}_0^\mathsf{T}\boldsymbol{A}_0\tilde{\boldsymbol{P}}_0) + \frac{t}{T}(\boldsymbol{P}_1^\mathsf{T}\boldsymbol{A}_1\boldsymbol{P}_1 - \tilde{\boldsymbol{P}}_1^\mathsf{T}\boldsymbol{A}_1\tilde{\boldsymbol{P}}_1)\|_F$$

$$\leq (1-\frac{t}{T})\|\boldsymbol{P}_0^\mathsf{T}\boldsymbol{A}_0(\boldsymbol{P}_0 - \tilde{\boldsymbol{P}}_0) + (\boldsymbol{P}_0 - \tilde{\boldsymbol{P}}_0)\boldsymbol{A}_0\tilde{\boldsymbol{P}}_0\|_F + \frac{t}{T}\|\boldsymbol{P}_1^\mathsf{T}\boldsymbol{A}_1(\boldsymbol{P}_1 - \tilde{\boldsymbol{P}}_1) + (\boldsymbol{P}_1 - \tilde{\boldsymbol{P}}_1)\boldsymbol{A}_1\tilde{\boldsymbol{P}}_1\|_F$$

$$\leq (1-\frac{t}{T})\Delta_{\boldsymbol{P}_0}\|\boldsymbol{A}_0\|_F(\|\boldsymbol{P}_0\|_F + \|\tilde{\boldsymbol{P}}_0\|_F) + \frac{t}{T}\Delta_{\boldsymbol{P}_1}\|\boldsymbol{A}_1\|_F(\|\boldsymbol{P}_1\|_F + \|\tilde{\boldsymbol{P}}_1\|_F)$$

Therefore, the approximation error can be bounded by

$$d_{\text{FGW}}^2(\mathcal{H}_t, \tilde{\mathcal{H}}_t) \leq (1-\frac{t}{T})\Delta_{\boldsymbol{P}_0}((1-\alpha)\|\boldsymbol{X}_0\|_F + \alpha\|\boldsymbol{A}_0\|_F(\|\boldsymbol{P}_0\|_F + \|\tilde{\boldsymbol{P}}_0\|_F)) + \frac{t}{T}\Delta_{\boldsymbol{P}_1}((1-\alpha)\|\boldsymbol{X}_1\|_F + \alpha\|\boldsymbol{A}_1\|_F(\|\boldsymbol{P}_1\|_F + \|\tilde{\boldsymbol{P}}_1\|_F))$$

Afterwards, we analyze the impact of the approximation error on the error bound. We first apply the triangle inequality on the FGW distance as

$$d_{\text{FGW}}(\tilde{\mathcal{H}}_t, \tilde{\mathcal{H}}_{t+1}) \leq d_{\text{FGW}}(\tilde{\mathcal{H}}_t, \mathcal{H}_t) + d_{\text{FGW}}(\mathcal{H}_t, \mathcal{H}_{t+1}) + d_{\text{FGW}}(\mathcal{H}_{t+1}, \tilde{\mathcal{H}}_{t+1})$$

Therefore, the error bound in Theorem 1 can be written as

$$\xi(f_T, G_0) \leq \text{Original bound} + 4C\delta_{\text{approx}}\sum_{t=1}^{T} d_{\text{FGW}}(\mathcal{H}_t, \mathcal{H}_{t+1}) + 4CT\delta_{\text{approx}}^2$$

Note that when $r \to |\mathcal{V}_1||\mathcal{V}_2|$, i.e., full rank, we have $\Delta_{\boldsymbol{P}_i} \to 0$, i.e., $\|\boldsymbol{A}_t - \tilde{\boldsymbol{A}}_t\|_F \to 0, \|\boldsymbol{X}_t - \tilde{\boldsymbol{X}}_t\|_F \to 0$. Therefore, we have $\delta_{\text{approx}} = max_t d_{\text{FGW}}(\mathcal{H}_t, \tilde{\mathcal{H}}_t) \to 0$ where approximation errors are eliminated. □

## B Algorithm

We first provide the detailed algorithm of the proposed GADGET in Algorithm 1, which generates the path for graph GDA.

---

**Algorithm 1** GADGET

---

1: **Input** source graph $\mathcal{G}_0 = (\mathcal{V}_0, \boldsymbol{A}_0, \boldsymbol{X}_0)$, target graph $\mathcal{G}_1 = (\mathcal{V}_1, \boldsymbol{A}_1, \boldsymbol{X}_1)$, number of stages $T$, marginals $\boldsymbol{\mu}_0, \boldsymbol{\mu}_1$, rank $r$, step size $\gamma$, lower bound $\alpha$, error threshold $\delta$.

2: Initialize transformation matrices $\boldsymbol{g}^{(0)} \in \Delta_r, \boldsymbol{Q}_0^{(0)} \in \Pi(\boldsymbol{\mu}_0, \boldsymbol{g}), \boldsymbol{Q}_1^{(0)} \in \Pi(\boldsymbol{\mu}_1, \boldsymbol{g})$;

3: Compute attribute distance matrix $\boldsymbol{M}(u, v) = \|\boldsymbol{X}_0(u) - \boldsymbol{X}_1(v)\|_2, \forall u \in \mathcal{V}_0, v \in \mathcal{V}_1$;

4: **for** $t = 0, 1, \ldots$ **do**

5:      $\boldsymbol{B}^{(t)} = -\alpha \boldsymbol{M} + 4(1 - \alpha)\boldsymbol{A}_0 \boldsymbol{Q}_0^{(t)}\text{diag}(1/\boldsymbol{g}^{(t)})\boldsymbol{Q}_1^{(t)^\top}\boldsymbol{A}_1$;

6:      $\boldsymbol{\xi}_1 = \exp\left(\gamma \boldsymbol{B}^{(t)}\boldsymbol{Q}_1^{(t)}\text{diag}(1/\boldsymbol{g}^{(t)})\right) \odot \boldsymbol{Q}_0^{(t)}$;

7:      $\boldsymbol{\xi}_2 = \exp\left(\gamma \boldsymbol{B}^{(t)^\top}\boldsymbol{Q}_0^{(t)}\text{diag}(1/\boldsymbol{g}^{(t)})\right) \odot \boldsymbol{Q}_1^{(t)^\top}$;

8:      $\boldsymbol{\xi}_3 = \exp\left(-\gamma\text{diag}\left(\boldsymbol{Q}_0^{(t)^\top}\boldsymbol{B}^{(t)}\boldsymbol{Q}_1^{(t)}\right)/\boldsymbol{g}^{(t)^2}\right) \odot \boldsymbol{g}^{(t)}$;

9:      $\boldsymbol{Q}_0^{(t+1)}, \boldsymbol{Q}_1^{(t+1)}, \boldsymbol{g}^{(t+1)} = \text{LR-Dykstra}(\boldsymbol{\xi}_1, \boldsymbol{\xi}_2, \boldsymbol{\xi}_3, \boldsymbol{\mu}_0, \boldsymbol{\mu}_1, \alpha, \delta)$ Scetbon et al. (2021);

10: **end for**

11: Normalize transformation matrices $\boldsymbol{P}_0 = \boldsymbol{Q}_0\text{diag}(1/\boldsymbol{g}), \boldsymbol{P}_1 = \boldsymbol{Q}_1\text{diag}(1/\boldsymbol{g})$;

12: Compute transformed adjacency matrices $\tilde{\boldsymbol{A}}_0 = \boldsymbol{P}_0^\top\boldsymbol{A}_0\boldsymbol{P}_0, \tilde{\boldsymbol{A}}_1 = \boldsymbol{P}_1^\top\boldsymbol{A}_1\boldsymbol{P}_1$;

13: Compute transformed attribute matrices $\tilde{\boldsymbol{X}}_0 = \boldsymbol{P}_0^\top\boldsymbol{X}_0, \tilde{\boldsymbol{X}}_1 = \boldsymbol{P}_1^\top\boldsymbol{X}_1$;

14: Compute transformed marginals $\tilde{\boldsymbol{\mu}}_0 = \boldsymbol{P}_0^\top\boldsymbol{\mu}_0, \tilde{\boldsymbol{\mu}}_1 = \boldsymbol{P}_1^\top\boldsymbol{\mu}_1$;

15: Generate intermediate graphs $\mathcal{H}_t := \left(\mathcal{V}_0 \otimes \mathcal{V}_1, \left(1 - \frac{t}{T}\right)\tilde{\boldsymbol{A}}_0 + \frac{t}{T}\tilde{\boldsymbol{A}}_1, \left(1 - \frac{t}{T}\right)\tilde{\boldsymbol{X}}_0 + \frac{t}{T}\tilde{\boldsymbol{X}}_1\right), \forall t = 1, 2, \ldots, T - 1$;

16: **return** path $\mathcal{H} = (\mathcal{H}_0, \mathcal{H}_1, \ldots, \mathcal{H}_T)$.

---

After obtaining the path by Algorithm 1, we can perform self-training along the path for GDA. The detailed algorithm is provided in Algorithm 2.

---

**Algorithm 2** Graph gradual domain adaptation

---

1: **Input** source graph $\mathcal{G}_0 = (\mathcal{V}_0, \boldsymbol{A}_0, \boldsymbol{X}_0)$, source node label $\boldsymbol{Y}_0$, target graph $\mathcal{G}_1 = (\mathcal{V}_1, \boldsymbol{A}_1, \boldsymbol{X}_1)$, number of stages $T$;

2: Generate path $\mathcal{H} = (\mathcal{H}_0, \mathcal{H}_1, \ldots \mathcal{H}_T)$ for graph GDA by GADGET in Algorithm 1

3: Set initial confidence score $\text{conf}_0 = \text{Unif}(|\mathcal{V}_0|)$

4: **for** $t = 0, 1, \ldots, T - 1$ **do**

5:      Train and adapt GNN model $f_t$ by $\arg\min_{f_\theta} \ell(\mathcal{H}_t, \mathcal{H}_{t+1}, \boldsymbol{Y}_{t+1}, \text{conf}_t)$;

6:      Obtain pseudo-labels by $\boldsymbol{Y}_{t+1} = f_t(\mathcal{H}_{t+1})$;

7:      Compute confidence score $\text{conf}_{t+1}$ on $\mathcal{H}_{t+1}$ by Eq. equation 15;

8: **end for**

9: **return** target GNN model $f_T$.

---

## C Additional Experiments

We provide additional experiments and analysis to better understand the proposed GADGET.

### C.1 Experiment Result Statistics

We provide more statistics on the benchmark results in Figure 2, and the statistics are shown in Table 1. We report the Average, Maximum and Minimum improvement of GADGET on direct adaptation with different DA methods and backbone GNNs. We also report the percentage of cases where GADGET outperforms (Positive) or underperforms (Negative) direct adaptation. It is shown that GADGET achieves positive average improvement on all datasets, with impressive maximum improvements of at least 9.83%. For cases where

Table 1: Statistics on experiment results. All number are reported in percentage (%).

| Dataset | Average | Max | Min | Positive | Negative |
|---|---|---|---|---|---|
| Airport | 6.77 | 26.30 | -1.75 | 94.40 | 5.56 |
| Social | 3.58 | 15.00 | -2.51 | 91.57 | 8.33 |
| Citation | 3.43 | 9.83 | -1.81 | 91.57 | 8.33 |
| CSBM | 36.51 | 48.00 | 16.67 | 100.0 | 0.00 |

GADGET fails, it still achieves comparable results with at most 2.51% degradation. However, as the columns Positive and Negative show, GADGET outperforms direct DA in over 90% cases, with only less than 9% cases with negative transfer.

## C.2 Computation Complexity Analysis

We analyze the time complexity of GADGET. Suppose we have source and target graphs with $\mathcal{O}(n)$ nodes, node feature dimension of $d$, and low-rank OT rank of $r$. The time complexity for path generation is $\mathcal{O}(Lndr+Ln^2r)$, where $L$ is the number of iterations in the low-rank OT algorithm in Algorithm 1. Besides, as gradual GDA involves repeated training along the path, an additional $\mathcal{O}(Tt_{\text{train}})$ complexity is needed, where $\mathcal{O}(t_{\text{train}})$ is the time complexity for training a GNN model. Therefore, the overall training complexity is $\mathcal{O}(Lndr + Ln^2r + Tt_{\text{train}})$, which is linear w.r.t. the feature dimension $d$ and the number of steps $T$, and quadratic w.r.t. the number of nodes $n$.

We also carry out experiments to analyze the run time w.r.t. the number of nodes $n$ with different ranks $r$, and the result is shown in Figure 8. It is shown that GADGET scales relatively well w.r.t. the number of nodes, exhibiting a sublinear scaling

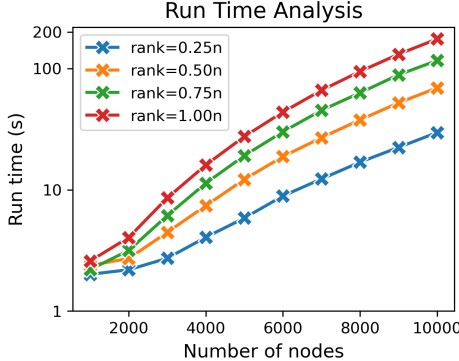

Figure 8: Run time analysis w.r.t. graph size. The y-axis is in the log scale.

of log(time) w.r.t. the number of nodes. Moreover, the computation can be further accelerated by reducing the rank. When $r$ is reduced from full-rank ($1.00n$) to low-rank ($0.25n$), the run time can be reduced from 175 seconds to 30 seconds on graphs with 10,000 nodes.

## C.3 Intermediate graphs.

We provide visualization results to understand the proposed graph GDA process, where the intermediate graphs between a 3-block graph and 2-block graph are shown in Figure 9. We observe a smooth transition from 3-block graph to 2-block graph with small shifts between two consecutive graphs.

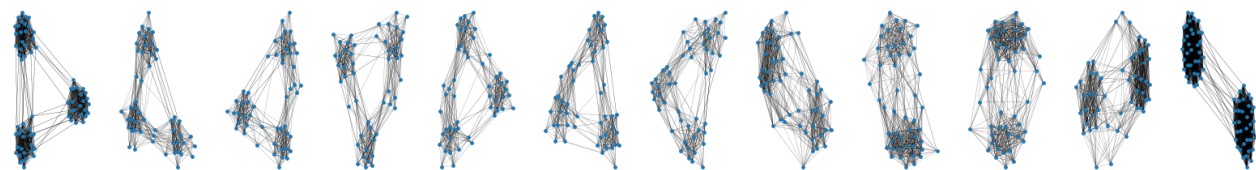

Figure 9: Visualization of the intermediate graphs.

## C.4 Pseudo-label confidence

To understand how entropy-based confidence facilitates self-training, we visualize the embedding spaces learned with and without entropy-based confidence, and the results are shown in Figure 10. It is shown that noisy pseudo-labels near the decision boundary are assigned with lower confidence, contributing less

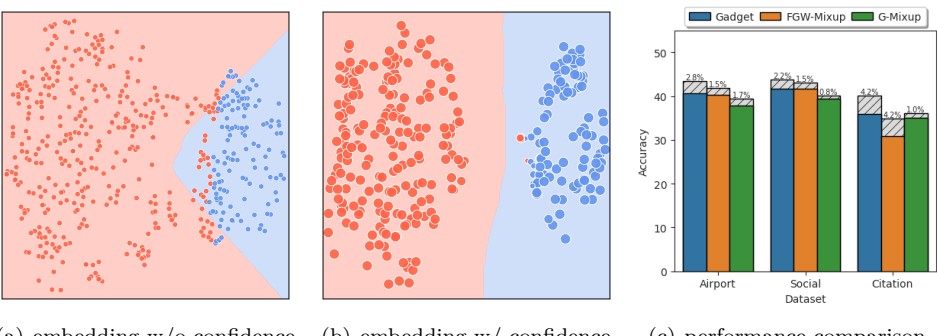

(a) embedding w/o confidence  (b) embedding w/ confidence  (c) performance comparison

Figure 10: Evaluation on pseudo-label quality. Larger marker size indicate more confident pseudo-label. (a) Embedding space w/o confidence; (b) Embedding space w/ confidence. (c) Performance comparison: we evaluate graph GDA guided by different paths w/ (hatched bars) and w/o (colored bars) confidence scores.

to self-training. In addition, we observe that the embedding space trained with confidence better separates different classes in the target domain, hence achieving better performance. Besides, we also quantitatively evaluate the universal benefits of entropy-based confidence by generating the intermediate graphs via different graph mixup methods Han et al. (2022); Ma et al. (2024).

# D   Reproducibility

## D.1   Datasets

We first introduce the datasets used in this paper, including three real-world datasets and three synthetic CSBM datasets, and the datasets statistics are provided in Table 2. For real-world datasets, we give a brief introduction as follows

- **Airport Ribeiro et al. (2017)** is a set of airport traffic networks, each of which is an unweighted, undirected network with nodes as airports and edges indicate the existence of commercial flights. Node labels indicate the level of activity of the corresponding airport. We use degree-bucking to generate one-hot node feature embeddings. The dataset includes three airports from *USA*, *Europe* and *Brazil*.

- **Citation Tang et al. (2008)** is a set of co-authorship networks, where nodes represent authors and an edge exists between two authors if they co-authored at least one publication. Node labels indicate the research domain of the author, including "Database", "Data mining", "Artificial intelligence", "Computer vision", "Information security" and "High performance computing". Node features are extracted from the paper content. The dataset includes two co-author networks from *ACM* and *DBLP*.

- **Social Li et al. (2015)** is a set of blog networks, where nodes represent bloggers and edges represent friendship. Node labels indicate the joining groups of the bloggers. Node features are extracted from blogger's self-description. The dataset includes two blog networks from BlogCatalog (*Blog1*) and Flickr (*Blog2*).

For synthetic datasets, we generate them based on the contextual block stochastic model (CSBM) Deshpande et al. (2018). In general, we consider a CSBM with two classes $\mathcal{C}_+ = \{v_i : y_i = +1\}$ and $\mathcal{C}_- = \{v_i : y_i = -1\}$, each with $\frac{N}{2}$ nodes. For a node $v_i$, the node attribute is independently sampled from a Gaussian distribution $\boldsymbol{x}_i \sim \mathcal{N}(\boldsymbol{\mu}_i, \boldsymbol{I})$. For nodes from class $\mathcal{C}_+$, we have $\boldsymbol{\mu}_i = \boldsymbol{\mu}_+$; and for nodes from class $\mathcal{C}_-$, we have $\boldsymbol{\mu}_i = \boldsymbol{\mu}_-$.

Table 2: Dataset statistics.

| Dataset | Domain | #node | #edge | #feat | #class |
|---------|--------|-------|-------|-------|--------|
| Airport | USA | 1,190 | 13,599 | 64 | 4 |
|         | Brazil | 131 | 1,038 | 64 | 4 |
|         | Europe | 399 | 6,193 | 64 | 4 |
| Citation | ACM | 7,410 | 14,728 | 7,537 | 6 |
|          | DCLP | 5,995 | 10,079 | 7,537 | 6 |
| Social | Blog1 | 2,300 | 34,621 | 8,189 | 6 |
|        | Blog2 | 2,896 | 55,284 | 8,189 | 6 |
| CSBM | Left | 500 | 5,154 | 64 | 2 |
|      | Right | 500 | 5,315 | 64 | 2 |
|      | Low | 500 | 2,673 | 64 | 2 |
|      | High | 500 | 10,302 | 64 | 2 |
|      | Homophily | 500 | 5,154 | 64 | 2 |
|      | Heterophily | 500 | 5,163 | 64 | 2 |

Each pair of nodes are connected with probability $p$ if they are from the same class, otherwise $q$. By varying the value of $\boldsymbol{\mu}_+, \boldsymbol{\mu}_-$, we can generate graphs with feature shifts. By varying the value of $p, q$, we can generate graphs with homophily shifts with homophily score as $h = \frac{p}{p+q}$, and degree shifts with average degree as $d = \frac{N(p+q)}{2}$. We provide more detailed description of generating the CSBM graphs as follows

- **CSBM-Attribute** is a set of CSBM graphs with attribute shifts. We generate two graphs with attributes shifted left (namely *Left*) and right (namely *Right*). We set the number of nodes as 500, homophily score as $h = 0.5$, average degree as 40, and feature dimension as 64. For node attributes, we set the $\boldsymbol{\mu}_+ = 0.6, \boldsymbol{\mu}_- = -0.4$ for *Right*, and $\boldsymbol{\mu}_+ = 0.4, \boldsymbol{\mu}_- = -0.6$ for *Left*.

- **CSBM-Degree** is a set of CSBM graphs with degree shifts. We generate two graphs with degree shifted high (namely *High*) and low (namely *Low*). We set the number of nodes as 500, homophily score as $h = 0.5$, feature dimension as 64, and features with $\boldsymbol{\mu}_+ = 0.5, \boldsymbol{\mu}_- = -0.5$. For node degree, we set $d = 80$ for *High* and $d = 20$ for *Low*.

- **CSBM-Homophily** is a set of CSBM graphs with homophily shifts. We generate two graphs with homophilic score (namely *Homophily*) and heterophilic score (namely *Heterophily*). We set the number of nodes as 500, average degree as 40, feature dimension as 64, and features with $\boldsymbol{\mu}_+ = 0.5, \boldsymbol{\mu}_- = -0.5$. For homophily score, we set the $h = 0.8$ for *Homophily*, and $h = 0.2$ for *Heterophily*.

### D.2 Pipeline

We focus on the unsupervised node classification task, where we have full access to the source graph, the source node labels, and the target graph during training. Our main experiments include two parts, including direct adaptation and GDA using GADGET. For direct adaptation, we perform directly adapt the source graph to target graph. For GDA, we first use GADGET to generate intermediate graphs, then gradually adapt along the path.

For path generation, we set the number of intermediate graphs as $T = 3$, and have all graphs uniformly distributed on the geodesic connecting source and target. We set $q = 2$ and $\alpha = 0.5$ for the FGW distance, and adopt uniform distributions $\text{Unif}(|\mathcal{V}_0|), \text{Unif}(|\mathcal{V}_1|)$ as the marginals.

For GNN models, we adopt light 2-layer GNNs with 8 hidden dimensions for smaller Airport and CSBM datasets, and heavier 3-layer GNNs with 16 hidden dimensions for larger Social and Citation datasets. We set the initial learning rate as $5 \times 10^{-2}$ and train the model for 1,000 epochs.

We implement the proposed method in Python and all backbone models based on PyTorch. For model training, all GNN models are trained on the Linux platform with an Intel Xeon Gold 6240R CPU and an NVIDIA Tesla V100 SXM2 GPU. We run all experiments for 5 times and report the average performance.

## E More Related Works

**Graph Domain Adaptation**  Graph DA transfers knowledge between graphs with different distributions and can be broadly categorized into data and model adaptation. Early graph DA methods drew inspiration from vision tasks by applying adversarial training to learn domain-invariant node embeddings Shen et al. (2020); Dai et al. (2022), analogous to DANN in images Ganin et al. (2016). Wu et al. (2020) introduced an unsupervised domain adaptive GCN that minimizes distribution discrepancy between graphs. Others exploit structural properties Wu et al. (2023); Guo et al. (2022), such as degree distribution differences Guo et al. (2022) and Subtree distance Wu et al. (2023). A hierarchical structure is further proposed by Shi et al. (2023a) to align graph structures hierarchically. The rapid progress in this area has led to dedicated benchmarks Shi et al. (2023b) and surveys Wu et al. (2024); Shi et al. (2024), consolidating GDA techniques. These studies consistently report that large distribution shifts between non-IID graph domains remain difficult to overcome, motivating novel solutions such as our OT-based geodesic approach for more effective cross-graph knowledge transfer.

**Gradual Domain Adaptation**  Gradual domain adaptation (GDA) addresses scenarios of extreme domain shifts by introducing a sequence of intermediate domains that smoothly connect the source to the target. Traditional methods in vision have instantiated the idea of GDA by generating intermediate feature spaces or image styles that interpolate between domains Gong et al. (2019); Hsu et al. (2020). For instance, DLOW Gong et al. (2019) learns a domain flow to progressively morph source images toward target appearance, and progressive adaptation techniques have improved object detection across environments Hsu et al. (2020). Recently, the theory of gradual adaptation has been formalized Kumar et al. (2020); Wang et al. (2022); Abnar et al. (2021); Chen & Chao (2021), where the benefits of intermediate distributions and optimal path have been studied. He et al. (2023) further provides generalization bounds proving the efficacy of gradual adaptation under certain conditions. On the algorithmic front, methods to construct or simulate intermediate domains have emerged. Sagawa & Hino (2022) leverages normalizing flows to synthesize a continuum of distributions bridging source and target, while Zhuang et al. (2024) employs a Wasserstein gradient flow to gradually transport source samples toward the target distribution. This gradual paradigm has only just begun to be explored for graph data, e.g., recent work suggests that interpolating graph distributions can significantly improve cross-graph transfer when direct adaptation fails due to a large shift. By viewing domain shift as a trajectory in a suitable metric space, one can effectively guide the model through intermediate graph domains, which is precisely the principle our FGW geodesic strategy instantiates.

**Graph Neural Networks**  Graph Neural Network (GNN) is a prominent approach for learning on graph-structured data, with wide applications in fields such as social network analysis Jing et al. (2024); Fu & He (2021); Yan et al. (2024a), bioinformatics Fu & He (2022); Xu et al. (2024b), information retrieval Wei et al. (2020); Li et al. (2024a;b); Liu et al. (2024c) and recommendation Liu et al. (2024b); Zeng et al. (2024b; 2025a;b); Liang et al. (2025); Yoo et al. (2023), and tasks like graph classification Xu et al. (2018); Lin et al. (2024b); Zheng et al. (2024), node classification Yan et al. (2024c); Liu et al. (2023b); Lin et al. (2024a); Xu et al. (2024a); Yan et al. (2023), link prediction Yan et al. (2022; 2024b), and time-series forecasting Lin et al. (2025; 2024c); Qiu et al. (2023); Wang et al. (2023). Foundational architectures such as GCN Kipf & Welling (2017), GraphSAGE Hamilton et al. (2017), and GAT Velickovic et al. (2017) introduced effective message-passing schemes to aggregate neighbor information, and subsequent variants have continuously pushed state-of-the-art performance. However, distribution shift poses a serious challenge to GNNs in practice: models trained on a source graph often degrade when applied to a different graph whose properties deviate significantly. This lack of robustness to domain change has prompted research into both graph domain generalization and graph domain adaptation. On the generalization side, methods inject regularization or data augmentation to make GNNs invariant to distribution changes such as graph mixup Ma et al. (2024); Zeng et al. (2024c); Zhou et al. (2024). On the adaptation side, numerous domain-adaptive GNN frameworks aim to transfer knowledge from a labeled source graph to an unlabeled target graph by aligning feature and

structural representations Shen et al. (2020); Dai et al. (2022); Liu et al. (2023a). Despite these advances, adapting GNNs to out-of-distribution graphs remains non-trivial, especially under large shifts. Besides, test-time adaptation on graphs has been studied recently Bao et al. (2024); Chen et al. (2022b); Jin et al. (2022); Zeng et al. (2026) where the GNN model is adapted at test time without re-accessing the source graph. However, existing graph DA methods implicitly assume a mild shift between the source and the target graph, while our work focuses on the more challenging setting where source and target graphs suffer from large shifts.

**Optimal Transport on Graphs** Optimal Transport (OT) provides a principled framework to compare and align distributions with geometric awareness, making it particularly well-suited for graph-structured data. OT-based methods have been used in graph alignment Xu et al. (2019); Zeng et al. (2023a; 2024a); Yu et al. (2025b;a), graph comparison Maretic et al. (2019); Titouan et al. (2019), and graph representation learning Kolouri et al. (2021); Vincent-Cuaz et al. (2021); Zeng et al. (2023b). The Gromov–Wasserstein (GW) distance Mémoli (2011); Peyré et al. (2016) enables comparison between graphs with different node sets and topologies, and defines a metric space where geodesics can be explicitly characterized Sturm (2012). Recent work Scetbon et al. (2022) further demonstrates how OT couplings can serve as transport maps that align and interpolate between graphs in this space. These advances provide the theoretical foundation for our work, which leverages Fused GW distances to construct geodesic paths between graph domains for GDA.

## F  Limitations and Future Directions

In this paper, we explore the idea of apply GDA for non-IID graph data to handle large graph shifts. We mainly focus on the unsupervised DA setting, with only one source domain and one target domain. Based on this limitation, we discuss possible directions and applications to further benefit and extend the current framework, including:

- `Multi-source graph GDA.` In this paper, we focus on the graph DA setting with one labeled source graph and unlabeled target graph. In real-world scenarios, we often have labeled information from multiple domains. Therefore, it would be beneficial to study multi-source graph GDA to leverage information from multiple source graphs.

- `Few-shot graph GDA.` In this paper, we focus on the unsupervised graph DA task where there is no label information for target samples. There may be cases where few target labels are available, and it would be beneficial to leverage such information into the graph GDA process. One possible solution is to leverage the graph mixup techniques Han et al. (2022); Ma et al. (2024); Zeng et al. (2024c) to generate pseudo-labels for intermediate nodes by the linear interpolation of source and target samples.

- `When to adapt.` While we mainly focus on how to best adapt the GNN model, an important question is when to adapt. For example, to what extent the domain shift is large enough to perform GDA? To what extent the domain shift is mild enough to perform direct adaptation or no adaptation. We believe that more powerful graph domain discrepancies such as the FGW distance provide solution to this problem.

- `Scalable GDA via Graph Coarsening.` To extend GADGET to massive-scale graphs (e.g., millions of nodes) where quadratic complexity is prohibitive, a promising future direction is Hierarchical Graph GDA. We plan to incorporate graph coarsening techniques (e.g., spectral clustering or edge contraction) to abstract the original graph into a smaller "super-node" graph. The FGW geodesic and transport plan can be efficiently computed on this coarsened level and then projected back to the original fine-grained graph. This "Coarsen-Align-Refine" strategy aims to reduce the effective number of nodes $n$ in the OT solver, potentially achieving near-linear time complexity while preserving global structural alignment.

