# OpenReview forum: "Pave Your Own Path: Graph Gradual Domain Adaptation on Fused Gromov-Wasserstein Geodesics"
_TMLR — Accepted by TMLR_

### Review · Reviewer_eCgC · 2025-11-04

**Summary Of Contributions:**

The paper tackles the graph Gradual Domain Adaptation (GDA) with Graph Neural Networks (GNN), handling large shifts between source and target graphs. In fact, as stated by the authors, here are the main contributions in this paper:
1) they extend GDA to non-IID graphs by leveraging the Fused Gromov-Wasserstein (FGW) distance;
2) they derive an error bound for graph GDA, highlighting the impact of the source GNN loss, the accumulated training error and the generalization error measured by length of the path;
3) they show that there exist an optimal path minimizing this error bound;
4) they provide an efficient way to generate intermediate graphs on this path;
5) they provide an extensive empirical evaluation of their method, showing improvements over existing DA methods on classification datasets.

Strengths:
- The paper is rather well written.
- The related works are discussed and the paper is clearly motivated.
- The claims are supported by theoretical and empirical evidences.

Weaknesses:
- The theoretical assumptions should be more commented.
- The computational cost of the method is not clearly discussed.
- The paper lacks some discussions on the limitations of the frameworks and future work directions.

**Audience:**

Yes

**Audience Explanation:**

I am not familiar with this literature, but the related works seem thoroughly discussed and the authors identified a hole in the literature regarding GDA. In fact, they propose to use FGW to handle non-IID graphs and provide some theoretical guarantees. Hence, in my opinion, the TMLR's audience should be interested in this paper.

**Claims And Evidence:**

Yes

**Claims Explanation:**

The authors support their framework by providing theoretical guarantees, namely the error bound and the optimal path. All of these are provided with proofs. Moreover, they provide an extensive empirical evaluation, including accuracy results on various classification datasets, domain shifts, hyperparameter and path quality analyses.

**Requested Changes:**

1) For the presentation, I would move the related works section after the introduction.
2) Could you please elaborate a little more about the assumptions you consider? First, could you comment assumptions A and B? Second, are all of these assumptions standard in the literature or do you introduce new ones?
3) Could you give more insights about the computational cost of the whole method? What is the complexity of generating the optimal path for instance? How does your method compare with respect to existing GDA methods in terms of computational cost?
4) Could you elaborate a little more about the limitations of your frameworks in your opinion and any research directions to alleviate them?

---

> ### Author Response · Authors · 2025-12-16
> **Response to Reviewer eCgC (1/2)**
>
> We thank the reviewer for the constructive feedback and comments, all of which are helpful to further improve our paper. In the following, we would like to provide point-to-point responses on **Weaknesses (W) and Requested Changes (C)**. We also have **revised our paper (colored in blue)** to accommodate these suggestions.
>
> > W1,C2: Justification on theoretical assumptions
>
> Thanks for the question. We would like to justify our assumptions as follows
>
> - **Assumption A & B: Stability of the GNN Model.** Both assumptions assume the smothness w.r.t. graph topology. Assumption A requires that the GNN model is stable: if two nodes share similar local neighborhoods, their embeddings should be similar. Assumption B implies that the model parameters are finite, which is naturally satisfied in practice through regularization techniques (e.g., weight decay). Both smoothness assumptions ensure GNNs' generalization and robustness. Previous works on graph DA [2,3] have also adopted similar assumptions to ensure the stability of GNN models.
>
> - **Assumption C & D: Smoothness of the Loss Landscape.** Both assumptions assume the smoothness w.r.t. predictions, i.e., the loss function does not change abruptly with small changes in model predictions. Intuitively, if the embeddings of a source node and a target node are close, their contribution to the adaptation loss should also be close. Assumption D (Holder continuity) specifically accommodates metrics defined with higher orders (e.g., $q=2$ in FGW). Previous works on gradual DA [4] have also adopted similar assumptions on the loss to relate target-source risk gaps to domain discrepancies.
>
> - **Assumption E & F: Consistency of Structural Operators.** Both assumptions assume the stability of readout, ensuring that the aggregation (Edge-level in Assumption E) and pooling (Graph-level in Assumption F) functions preserve the proximity of node embeddings. It guarantees that errors in node-level representations do not explode when aggregated into edge or graph representations. Previous works on GNN stability [5] have also adopted similar assumptions on structural operators to guarantee that node-level perturbations do not amplify at higher structural levels.
>
> We have included the above discussions in the revised version to make the theory more accessible.
>
> > W2, C3: Discussion on computational cost
>
> Thanks for the question. We would like to provide the following analysis on computational cost.
>
> >> W2.1: Overall computational analysis
>
> As discussed in Appendix C.2, the overall training complexity is $O(Lndr+Ln^2r+Tt_{train})$, which is linear w.r.t. the feature dimension $d$ and the number of steps $T$, and quadratic w.r.t. the number of nodes $n$.
>
> Specifically, regarding the cost of generating the optimal path:
> - The complexity is dominated by the low-rank OT solver (Algorithm 1), which is $O(Lndr + Ln^2r)$.
> - The term $Lndr$ arises from the matrix multiplications involving node features (Line 6, 7 in Algorithm 1).
> - The term $Ln^2r$ arises from the structure mapping steps (Line 9 in Algorithm 1).
> - Crucially, this complexity is linear with respect to the feature dimension $d$ and quadratic with respect to the number of nodes $n$ (assuming $r \ll n$).
>
> The computational analysis is further supported by the empirical analysis in Figure 5 and Figure 7, showing how the run time changes with different $T$ and $r$.
>
> >> W2.3 Comparison to baseline GDA methods in computational cost.
>
> First, we would like to clarify that our work is the first work on gradual domain adaptation (GDA) on structured data like graphs.
> Previous works [3, 5-7] **predominantly focus on Euclidean data like images, while ignoring the structure information.** However, we can categorize the comparison as follows:
> - vs. Predefined Path Methods [3, 5, 6]: These methods (e.g., rotating images or changing lighting) assume intermediate domains are given or trivially constructible. While they have near-zero generation cost, they cannot handle non-IID graph structure shifts where no natural intermediate domains exist.
> - vs. Generative Methods [7], which generates intermediate samples on Wasserstein Geodesics.
>     - Baseline Cost: Standard Wasserstein geodesic generation typically involves solving exact OT, which has a cubic complexity of $O(n^3 \log n)$.
>     - Our Advantage: By leveraging the low-rank approximation in the FGW space, GADGET reduces the complexity to $O(n^2 r)$. Since graphs often possess sparse and low-rank structures, we can adopt a small rank $r \ll n$.
>     - Result: As shown in Figure 5(b) 4, a small rank $r$ achieves performance comparable to full-rank but with significantly lower computational cost, making GADGET even more scalable than direct adaptations of cubic-complexity generative GDA methods.

---

> ### Author Response · Authors · 2025-12-16
> **Response to Reviewer eCgC (2/2)**
>
> > W3, C4: Discussions on limitations and future works
>
> Thanks for the question. We discussed the limitations and future works in Appendix E.
>
> For limitations, we include
> - **Setting Scope**: our work only consider settings with single source and target domain. It does not yet strictly cover multi-source scenarios where labeled data from diverse domains are available.
> - **Scalability**: Though we adopt low-rank approximation for speedup, the path generation is quadratic complexity w.r.t. the graph size $n$. This may still pose challenges for extremely large-scale graphs (e.g., millions of nodes).
>
> For future directions, we include
> - **Multi-source graph GDA** to embrace more diverse signals from multiple domains.
> - **Few-shot graph GDA** where we have a limited number of target labels to help the adaptation process.
> - **Adaptive GDA**, where we first measure the domain shift level and adaptively select different GDA strategies to achieve better effectiveness and efficiency.
> - **Scalable GDA via Graph Coarsening**: To extend GADGET to massive-scale graphs (e.g., millions of nodes) where quadratic complexity is prohibitive, a promising future direction is Hierarchical Graph GDA. We plan to incorporate graph coarsening techniques (e.g., spectral clustering or edge contraction) to abstract the original graph into a smaller "super-node" graph. The FGW geodesic and transport plan can be efficiently computed on this coarsened level and then projected back to the original fine-grained graph. This "Coarsen-Align-Refine" strategy aims to reduce the effective number of nodes $n$ in the OT solver, potentially achieving near-linear time complexity while preserving global structural alignment.
>
> > C1: Move related works section after introduction
>
> Thank you for the suggestion. We have placed the related work section immediately after the introduction to better equip readers with the necessary background.
>
> > References
>
> [1] You, Yuning, et al. "Graph domain adaptation via theory-grounded spectral regularization." The eleventh international conference on learning representations. 2023.
>
> [2] Arghal, Raghu, Eric Lei, and Shirin Saeedi Bidokhti. "Robust graph neural networks via probabilistic lipschitz constraints." Learning for Dynamics and Control Conference. PMLR, 2022.
>
> [3] Wang, Haoxiang, Bo Li, and Han Zhao. "Understanding gradual domain adaptation: Improved analysis, optimal path and beyond." International Conference on Machine Learning. PMLR, 2022.
>
> [4] Davidson, Yair, and Nadav Dym. "On the Hölder Stability of Multiset and Graph Neural Networks." The Thirteenth International Conference on Learning Representations.
>
> [5] Kumar, Ananya, Tengyu Ma, and Percy Liang. "Understanding self-training for gradual domain adaptation." International conference on machine learning. PMLR, 2020.
>
> [6] Chen, Hong-You, and Wei-Lun Chao. "Gradual domain adaptation without indexed intermediate domains." Advances in neural information processing systems 34 (2021): 8201-8214.
>
> [7] He, Yifei, et al. "Gradual domain adaptation: Theory and algorithms." Journal of Machine Learning Research 25.361 (2024): 1-40.

---

### Review · Reviewer_83tt · 2025-11-10

**Summary Of Contributions:**

This paper introduces GADGET, the first framework for gradual domain adaptation (GDA) on non-IID graph data, designed to address the challenge of large distribution shifts where conventional graph DA methods often fail. The authors establish a rigorous theoretical foundation by adopting the Fused Gromov-Wasserstein (FGW) distance to measure discrepancy between graphs, which uniquely accounts for both node attributes and graph structure. Based on this, they derive a novel error bound for graph GDA, revealing that the target domain error is proportional to the length of the adaptation path. Motivated by this bound, the paper proves that the optimal path is the FGW geodesic and proposes an efficient practical algorithm to generate intermediate graphs along this path. The framework is designed as a versatile wrapper that can be seamlessly integrated with existing graph DA methods, and its effectiveness is demonstrated through extensive experiments, showing significant performance improvements on various real-world datasets.

**Audience:**

Yes

**Audience Explanation:**

The findings of this paper would be of interest to the TMLR audience, particularly those working in graph machine learning, transfer learning, and robust AI.

**Broader Impact Concerns:**

No concerns.

**Claims And Evidence:**

No

**Claims Explanation:**

The evidence has a notable weakness, creating a gap between the convincing theory and the convincing experiments. The theoretical proofs of optimality rely on the true FGW geodesic, but for practical scalability, the paper's algorithm implements a low-rank approximation. The review points out that the submission lacks a theoretical analysis of the error introduced by this approximation. Consequently, while the algorithm is empirically effective, the evidence that the generated path is a close or sufficient approximation of the theoretically optimal path is not provided, making this part of the claim less supported. Furthermore, the review notes the algorithm's novelty is "somewhat limited" as it is a direct adaptation of a prior method, which slightly tempers the claim of a novel generation technique.

**Requested Changes:**

1. **Clarify Algorithmic Novelty and Positioning:** The path generation algorithm presented in Section 4 is a direct adaptation of the method from Zeng et al. (2024). The current "Remark" on page 8 is insufficient. Please revise the main text in Section 4 to explicitly state that the algorithm adapts this prior work for the FGW space and the novel task of unsupervised node-level GDA. This will more accurately frame the paper's core contribution, which is the theoretical framework and problem formulation for gradual adaptation on graphs, rather than a new optimization algorithm.
2. **Address the Gap Between Theory and Practice:** There is a significant disconnect between the theoretical analysis, which relies on the exact FGW geodesic (Theorems 1-3), and the practical algorithm, which uses a low-rank approximation for scalability (Eq. 14). The paper currently lacks any analysis of the error introduced by this approximation.
3. **Strengthen Justification for Theoretical Assumptions:** The mathematical assumptions in Section 3.2 need better intuition and justification to make the theoretical claims more convincing and accessible.

---

> ### Author Response · Authors · 2025-12-16
> **Response to Reviewer 83tt (1/2)**
>
> We thank the reviewer for the constructive feedback and comments, all of which are helpful to further improve our paper. In the following, we would like to provide point-to-point responses on **Weaknesses (W) and Requested Changes (C)**. We also have **revised our paper (colored in blue)** to accommodate these suggestions.
>
> > C1: Clarification on algorithmic novelty and positioning.
>
> Thanks for the suggestion. We acknowledge that the path generation algorithm serves as a tool to realize our proposed framework, and our core contributions lie in the **theoretical formulation of graph GDA, the error bound derivation, and the identification of the FGW geodesic as the optimal path**, rather than the optimization solver itself.
>
> Per your suggestion, we have revised Section 4 to explicitly position the path generation algorithm as an adaptation of the solver from [1], extended to the Fused Gromov-Wasserstein (FGW) space for the unsupervised GDA task. Specifically:
>
> - Better Positioning: We revised the opening of Section 4 to state: "To address the lack of path in graph learning tasks, we employ an efficient low-rank OT algorithm adapted from [1] to generate intermediate graphs on the FGW geodesics."
>
> - Detailed Context: We revised the remark paragraph in Section 4. This explicitly clarifies that while the solver logic is similar, our work incorporates node attributes (FGW) and adapts the pipeline for unsupervised domain adaptation (via self-training) , distinct from the supervised graph mixup task in prior work.
>
>
>
> > C2: Gap betwen theory and practice.
>
> Thanks for the insightgul suggestion! To address this, we have derived a rigorous error bound for the approximation gap and analyzed its impact on the final generalization error. The analysis proceeds in three steps: (1) Bounding the distance between exact and approximate graphs using the coupling error; (2) Justifying the convergence of the coupling error; (3) Incorporating this gap into the final error bound.
>
> >> C2.1:  Theoretical analysis on the effects of geodesic approximation.
>
> Let $H$ be the graph on the exact geodesic and $\tilde{H}$ be the graph on the approximate path with rank-$r$. We use $P$ to denote the optimal coupling and $\tilde{P}$ to denote the approximated low-rank coupling. We denote $\Delta\_{P}=||P-\tilde{P}||\_F$. We denote $\delta\_{approx} =\max_t d\_{FGW}(H\_t, \tilde{H}\_t)$ as the maximum approximation error between the exact geodesic graph $H_t$ (full-rank) and approximated geodesic graph $\tilde{H}_t$
>
> **Step 1: Bounding the approximation gap $\delta_{approx}$.**
>
> When considering a naive identity coupling (i.e., the $i$-th node in $H_t$ align with the $i$-th node in $\tilde{H}_{t}$) between $H_t$ and $\tilde{H}_t$, it induces an upper bound of the FGW distacne as
>
> $d^2\_{FGW}(H\_t,\tilde{H}\_t)\leq (1-\alpha)||X\_t-\tilde{X}\_t||\_F + \alpha||A\_t-\tilde{A}\_t||\_F$.
>
> According to the transformation in Eq.(13), we can write feature error at step $t$ as
>
> $||X\_t-\tilde{X}\_t||\_F = ||(1-\frac{t}{T})(P\_0^T-\tilde{P}\_0^T)X\_0+ \frac{t}{T}(P\_1^T-\tilde{P}\_1^T)X\_1||_F\leq(1-\frac{t}{T})\Delta\_{P\_0}||X\_0||_F + \frac{t}{T}\Delta\_{P\_1}||X\_1||_F$
>
> Similarly, we can write the structure error at step $t$ as
>
> $||A_t-\tilde{A}_t||_F=||(1-\frac{t}{T})(P_0^TA_0P_0-\tilde{P}_0^TA_0\tilde{P}_0) + \frac{t}{T}(P_1^TA_1P_1-\tilde{P}_1^TA_1\tilde{P}_1)||_F$.
>
> We can further expand $||P\_i^TA\_iP-\tilde{P}\_i^TA\_i\tilde{P}\_i||_F=||P\_i^TA\_i(P\_i-\tilde{P}\_i) + (P\_i-\tilde{P}\_i)A\_i\tilde{P}\_i||\_F\leq \Delta\_{P\_i}||A\_i||(||P\_i||_F+||\tilde{P}\_i||\_F)$. Thereafter, we can write the bound for structure error as
>
> $||A\_t-\tilde{A}\_t||_F \leq (1-\frac{t}{T})\Delta\_{P_0}||A\_0||_F(||P\_0||\_F+||\tilde{P}\_0||\_F)+\frac{t}{T}\Delta\_{P\_1}||A\_1||\_F(||P\_1||\_F+||\tilde{P}\_1||\_F)$.
>
> Therefore, we bound the approximation error by
> $d^2\_{FGW}(H\_t,\tilde{H}\_{t})\leq (1-\frac{t}{T})\Delta\_{P\_0}((1-\alpha)||X\_0||\_F+\alpha||A\_0||\_F(||P\_0||\_F+||\tilde{P}\_0||\_F)) + \frac{t}{T}\Delta\_{P\_1}((1-\alpha)||X\_1||\_F+\alpha||A\_1||\_F(||P\_1||\_F+||\tilde{P}\_1||\_F))$
>
> **Step 2: Impact on the error bound**
> According to Theorem 1, the approximation error will increase the length of the path, hence increasing the error bound.
>
> Formally, by adopting the triangle inequality, we have $d\_{FGW}(\tilde{H}\_t,\tilde{H}\_{t+1}) \leq d\_{FGW}(\tilde{H}\_t,H\_t) + d\_{FGW}(H\_t,H\_{t+1}) + d\_{FGW}(H\_{t+1},\tilde{H}\_{t+1})$.
>
> Therefore, the error bound in Theorem 1 can be written as
> $\xi(f\_T,G\_0)\leq \text{Original bound} + 4C\delta\_{approx}\sum_{t=1}^Td\_{FGW}(H\_t,H\_{t+1}) + 4CT\delta\_{approx}^2$.
>
> **Note that when $r\to n\_1n\_2$, i.e., full rank, we have $\Delta\_{P_i}\to 0$, i.e., $||A\_t-\tilde{A}\_t||\_F\to 0, ||X\_t-\tilde{X}\_t||\_F\to 0$. Therefore, we have $\delta\_{approx}=max\_t d\_{FGW}(H\_t,\tilde{H}\_t)\to 0$ where approximation errors are eliminated**.
>
> (continued on next page)

---

> ### Author Response · Authors · 2025-12-16
> **Response to Reviewer 83tt (2/2)**
>
> >> C2.2: Empirical analysis
>
> Besides, we provide empirical analysis in our paper to justify the effectiveness of such low-rank approximation in both scalability and performance.
>
> First, we validate the empirical benefit of low-rank approximation. In Figure 5(b), we study the model performance (bars) and running time (curves) under different ranks, and it is shown that a relatively small rank $r$ can obtain **relatively good results with siginifcantly less run time**. Besides, in Appendix C.2, we provide a comprehensive complexity analysis, showing that adopting the low rank formulation induces a time complexity quadratic w.r.t. $n$ and linear w.r.t. $r$.
>
> Second, we evaluate the approximation error of the low-rank approximation. In Figure 6, we evaluate the geodesic property of the generated path, which shows that the ratio between the distances of two intermediate domains $d_{FGW}(G_{\lambda_0}, G_{\lambda_1})/d_{FGW}(G_0, G_1)$ is closely correlated with the difference between the interpolation ratio $|\lambda_0-\lambda_1|$, with a Pearson coefficient of over 0.98.
>
>
> > C3: Justification on theoretical assumptions
>
> Thanks for the question. We would like to justify our assumptions as follows
>
> - **Assumption A & B: Stability of the GNN Model.** Both assumptions assume the smoothness w.r.t. graph topology. Assumption A requires that the GNN model is stable: if two nodes share similar local neighborhoods, their embeddings should be similar. Assumption B implies that the model parameters are finite, which is naturally satisfied in practice through regularization techniques (e.g., weight decay). Both smoothness assumptions ensure GNNs' generalization and robustness. Previous works on graph DA [2,3] have also adopted similar assumptions to ensure the stability of GNN models.
>
> - **Assumption C & D: Smoothness of the Loss Landscape.** Both assumptions assume the smoothness w.r.t. predictions, i.e., the loss function does not change abruptly with small changes in model predictions. Intuitively, if the embeddings of a source node and a target node are close, their contribution to the adaptation loss should also be close. Assumption D (Holder continuity) specifically accommodates metrics defined with higher orders (e.g., $q=2$ in FGW). Previous works on gradual DA [4] have also adopted similar assumptions on the loss to relate target-source risk gaps to domain discrepancies.
>
> - **Assumption E & F: Consistency of Structural Operators.** Both assumptions assume the stability of readout, ensuring that the aggregation (Edge-level in Assumption E) and pooling (Graph-level in Assumption F) functions preserve the proximity of node embeddings. It guarantees that errors in node-level representations do not explode when aggregated into edge or graph representations. Previous works on GNN stability [5] have also adopted similar assumptions on structural operators to guarantee that node-level perturbations do not amplify at higher structural levels.
>
> We have included the above discussions in the revised version to make the theory more accessible.
>
>
> > References
>
> [1] Zeng, Zhichen, et al. "Graph mixup on approximate gromov–wasserstein geodesics." Forty-first International Conference on Machine Learning. 2024.
>
> [2] You, Yuning, et al. "Graph domain adaptation via theory-grounded spectral regularization." The eleventh international conference on learning representations. 2023.
>
> [3] Arghal, Raghu, Eric Lei, and Shirin Saeedi Bidokhti. "Robust graph neural networks via probabilistic lipschitz constraints." Learning for Dynamics and Control Conference. PMLR, 2022.
>
> [4] Wang, Haoxiang, Bo Li, and Han Zhao. "Understanding gradual domain adaptation: Improved analysis, optimal path and beyond." International Conference on Machine Learning. PMLR, 2022.
>
> [5] Davidson, Yair, and Nadav Dym. "On the Hölder Stability of Multiset and Graph Neural Networks." The Thirteenth International Conference on Learning Representations.

---

### Review · Reviewer_DNXZ · 2025-12-05

**Summary Of Contributions:**

The authors propose a new method called GADGET. This method is based on Gradual Domain Adaptation and Graph Neural Networks. It learns a mapping according to the Fused Gromov–Wasserstein distance and, instead of solving a single hard domain-adaptation problem, it solves a sequence of small ones along the geodesic of this mapping. The method is supported by an interesting theoretical study that provides insights into its behavior.
Strengths:
- The paper is easy to follow, and apart from the figures, everything is clear.
- The new method is supported by a theoretical study that provides insight into how many steps are optimal.
- The authors present interesting experiments and compare their approach with a variety of methods and architectures.

Weaknesses:
- It is difficult to distinguish the domains in Figure 1; the crosses and circles are not easily recognizable.
- The same issue appears in Figure 3.
- There is no discussion of why GADGET performs better on csbm but worse on airport, social, and media. Similarly, there is no intuition for why GADGET fails to improve the score for some methods. Could you provide a potential failure case to clarify when GADGET is or is not useful? For example, when GADGET fails, do we effectively end up with T = 1 in your formulation?
It is unclear how GADGET is applied across all baselines, given the diversity of methods: shallow techniques like CORAL, deep methods like AdaGCN, and already-gradual DA methods like GRADE. How is GADGET combined with GRADE, for instance? Is GRADE simply the specific case where alpha = 0?
- There is no discussion of the method’s limitations. Up to what graph size can GADGET be used? Solving FGW can be time-consuming when the number of nodes is large. What is the proposed solution? A larger dataset could also strengthen the experimental validation.

**Additional Comments:**

In figure 5, you are plotting the time. Does it refers to the total training of you method ? 1 sec seems very fast.

**Audience:**

Yes

**Audience Explanation:**

Yes, GNN is becoming a big field in machine learning nowadays and this paper brings a new method to tackle efficiently the shift between graph.

**Broader Impact Concerns:**

No concern

**Claims And Evidence:**

Yes

**Claims Explanation:**

The authors claim that DA method for graph was supposing mild shift when real datasets suffer from larger shift.
The theoretical study shows that the performance on the target is bounded by the shift between the two graph and the experiments show that limiting this shift with GDA brings increase in the performance.

**Requested Changes:**

- Add study of parameter alpha
- Add limit of the method in the conclusion.

---

> ### Author Response · Authors · 2025-12-16
> **Response to Reviewer DNXZ (1/3)**
>
> We thank the reviewer for the constructive feedback and comments, all of which are helpful to further improve our paper. In the following, we would like to provide point-to-point responses on **Weaknesses (W) and Requested Changes (C)**. We also have **revised our paper (colored in blue)** to accommodate these suggestions.
>
> > W1, W2: Clarity of Figures 1 and 3
>
> Sorry for the confusion, we have revised the two figures with with larger marker sizes to make them more clear.
>
> > W3: Discussions on failure cases
>
> Thanks for the insightful question. We have added the following discussions to Section 5.2.
>
> As Theorem 1 indicates, the error bound depends on three terms: (1) source GNN performance, (2) accumulated training error, and (3) generalization error. And we categorize possible failures into three categories.
>
>
> - **Mild Shifts (Dominant Training Error)**: When the domain shift is mild (small $d_{FGW}$), the Generalization Error (Term 3) is already negligible. In this case, direct adaptation ($T=1$) is sufficient. Forcing a gradual process ($T>1$) introduces unnecessary self-training steps, leading to an increase in Accumulated Training Error (Term 2) that outweighs the marginal gain in generalization.
> - **Weak Base Model (noisy pseudo labels)**: The Accumulated Training Error is proportional to $\delta$ (the error at each adaptation step). If the backbone GNN or the baseline DA method is too weak, it may produce noisy pseudo labels that misguide the gradual DA process. This noise propagates and amplifies through the self-training steps, misguiding the adaptation process and leading to negative transfer.
> - **Extreme Shifts**: When the shift is extremely large, the Generalization Error is massive. To reduce this error, Theorem 2 suggests we need a large number of steps $T$. However, since the Accumulated Training Error grows linearly with $T$ ($C_f \cdot \delta T$), there exists a "break-even point.": If the shift is so extreme that the $T$ required to bridge the domain gap introduces noise that exceeds the capacity of the model, GADGET may fail to recover accuracy.
>
> >> W3.1: Why GADGET is more effective on CSBM than others?
>
> The CSBM datasets were synthetically generated to **exhibit large distribution shifts** (e.g., flipping feature means, drastically changing homophily). Under such large shifts, the generalization error dominates. GADGET significantly reduces this error by decomposing the large shift into small steps, yielding massive gains (up to 48%).
>
> For real-world datasets, like Airport and Citation, the shifts are still large but less extreme. Under such settings, the benefit of geodesic path decomposition is less significant, while the accumulated training error (noise from self-training pseudo-labels) may outweigh the benefits. This aligns with our observation in Figure 4, where GADGET's advantage diminishes as the shift level decreases.
>
> >> W3.2: Why GADGET fails to improve some methods?
>
> As discussed before, there are two possible cases:
>
> - Distribution shift is mild, hence performing direct DA is good enough and performing gradual DA may introduce accumulated training error that degrade the performance. For example, this corresponds to cases like GCN+AdaGCN on Blog1-Blog2 (Figure 2c) and GCN+CORAL on ACM-DBLP (Figure 2d).
> - Baseline methods are too weak to provide reliable pseudo labels. And incorrect pseudo labels may misguide the GDA performance. For example, this corresponds to cases like GCN+MMD on Brazil-Europe in Figure 2b and GCN+CORAL on Europe-USA in Figure 2b.
>
>
> >> W3.3: when fails, does it effectively reduce to $T=1$?
>
> As discussed in W3.2, the failure can be attributed to two possible causes. When the distribution shift is mild, such failure can be efffectively addressed by setting $T=1$. However, when the base model is too weak, we may want to increase the value of $T$ to ensure per-step pseudo lanel quality.
>
> >> W3.4: How is GADGET combined with different baselines?
>
> Thanks for the question. We have added clarifications in Sec 5.1. In general, we combine GADGET with different baselines by treating the baseline method as the solver for the adaptation step between consecutive graphs along the geodesic path.
>
> Specifically, for a path $\mathcal{H}_0, \mathcal{H}_1, \dots, \mathcal{H}_T$ generated by GADGET, we instantiate the self-training step with different graph DA baselines as follows:
> - Sequential Adaptation: At step $t$, current graph $H_t$ (with pseudo-labels from previous model $f_{t-1}$) is used as the "source" and the next graph $\mathcal{H}_{t+1}$ as the "target".
> - Loss Instantiation: We may adopt different adaptation loss $\ell$ of the chosen baseline in Equation (2) for training.
> - Model Update: The baseline method is then tasked with adapting the model from $\mathcal{H}\_t$ to $\mathcal{H}\_{t+1}$.
>
> (continued on next page)

---

> ### Author Response · Authors · 2025-12-16
> **Response to Reviewer DNXZ (2/3)**
>
> >> W3.5: Is GRADE a specific case with $\alpha=0$?
>
> Thanks for the question. We would like to clarity the difference between GADGET and GRADE.
>
> - **Different Mathematical Metrics**: GRADE minimizes the Graph Subtree Discrepancy (GSD), which is the average of subtree distances at different levels, capturing structure information by aggregating local neighborhood information into subtree representations. However, we adopt the FGW distance, which measures the discrepancy between all node pairs, capturing structure information via the underlying geometry cost.
> - **Direct vs. Gradual adaptation**: GRADE is an direct adaptation with one step, while GADGET is a gradual DA framework with multiple steps. In fact, GRADE serves as the baseline adaptation method for GADGET, where we can perform DA between two consecutive graphs on the path generated by GADGET.
>
> Besides, we would like to clarify that GRADE is different from GADGET with $\alpha=0$. When $\alpha=0$, FGW distance degrades to wasserstein distance that ignores the graph structure. Therefore, as we stated in Appendxi D.2, we adopt $\alpha=0.5$ to achieve a good balance between node features and graph structure.
>
> > W4, C2: Discussion on limitations (scalability, larger datasets)
>
> Thanks for the question. We discussed the limitations and future works in Appendix E.
>
> For limitations, we include
> - **Setting Scope**: our work only consider settings with single source and target domain. It does not yet strictly cover multi-source scenarios where labeled data from diverse domains are available.
> - **Scalability**: Though we adopt low-rank approximation for speedup, the path generation is quadratic complexity w.r.t. the graph size $n$. This may still pose challenges for extremely large-scale graphs (e.g., millions of nodes).
>
> For future directions, we include
> - **Multi-source graph GDA** to embrace more diverse signals from multiple domains.
> - **Few-shot graph GDA** where we have a limited number of target labels to help the adaptation process.
> - **Adaptive GDA**, where we first measure the domain shift level and adaptively select different GDA strategies to achieve better effectiveness and efficiency.
> - **Scalable GDA via Graph Coarsening**: To extend GADGET to massive-scale graphs (e.g., millions of nodes) where quadratic complexity is prohibitive, a promising future direction is Hierarchical Graph GDA. We plan to incorporate graph coarsening techniques (e.g., spectral clustering or edge contraction) to abstract the original graph into a smaller "super-node" graph. The FGW geodesic and transport plan can be efficiently computed on this coarsened level and then projected back to the original fine-grained graph. This "Coarsen-Align-Refine" strategy aims to reduce the effective number of nodes $n$ in the OT solver, potentially achieving near-linear time complexity while preserving global structural alignment.
>
> > C1: Parameter study on $\alpha$.
>
> Thanks for the question. We have conducted additional study on the selection of $\alpha$ on Airport dataset. We report the average performance across different domain pairs.
>
> | $\alpha$ | ERM   | AdaGCN | GRADE |
> |-------|-------|--------|-------|
> |  0.00 | 35.13 |  41.38 | 40.58 |
> |  0.25 | 39.65 |  42.53 | 41.42 |
> |  0.50 | 40.63 |  43.53 | 42.95 |
> |  0.75 | 40.87 |  42.75 | 40.98 |
> |  1.00 | 38.15 |  42.50 | 40.42 |
>
> As the results shown above, we observe that
> - The performance is robust to different selections with in $\alpha\in(0,1)$.
> - When $\alpha=0$, distance degrades to Wasserstein distance considering feature only, and when $\alpha=1$, distance degrades to GW distance considering structure only. Both senarios lead to performance degradation. This validates that we should consider both feature and structure when constructing the path.
>
> We have included these studies in Appendix C.5 to make the evaluation more comprehensive.
>
> > W4: Time complexity and scalability.
>
> Thanks for the question.
>
> First, we would like to referto Appendix C.2, where we analyze the time complexity of GADGET to be linear w.r.t. the feature dimension $d$ and the number of steps $T$, and quadratic w.r.t. the number of nodes $n$.
>
> Besides, we provide the empirical run time of GADGET on graphs up to size of 10,000. It is shown that Gadget scales relatively well w.r.t. the number of nodes, exhibiting a sublinear scaling of log(time) w.r.t. the number of nodes. Also, as indicated in Figure 5(b) and 7, we can always to choose a small rank $r$ to achieve good scalability.
>
> In addition, we report the model performance on large CSBM dataset with 10,000 nodes in the table below. It is shown that GADGET consistent enhances baseline method performance.
>
> |Method| Homophily| Degree| Attribut|
> |--------------|------------|------------|------------|
> |ERM          | 59.50±2.40 | 64.90±3.39 | 66.75±0.64 |
> |ERM+GADGET   | 94.85±0.78 | 93.50±4.10 | 97.45±1.06 |
> |GRADE        | 65.40±4.67 | 73.55±1.34 | 75.65±0.64 |
> |GRADE+GADGET|96.00±4.10|94.70±4.38|98.10±0.99|

---

> ### Author Response · Authors · 2025-12-16
> **Response to Reviewer DNXZ (3/3)**
>
> > Additional Comment: Clarification on time in Figure 5
>
> Sorry for the confusion.
> To avoid ambiguity, we clarify the notion of time used in Figure 5. Subfigure (a) reports the training time of the downstream GNN as a function of the number of adaptation steps $T$, while subfigure (b) reports the path generation time as a function of the rank $r$ used in the low-rank FGW approximation. This design reflects that $T$ primarily influences the cost of model training, whereas $r$ mainly affects the complexity of geodesic path generation; the $y$-axes of both panels are labeled accordingly.

---

### Decision · Action_Editor_KKRA · 2026-01-30

**Recommendation:** Accept as is

**Audience:**

Yes

**Audience Explanation:**

The topic is relevant to the TMLR audience, especially for readers interested in graph domain adaptation. In particular, I think the theoretical development for gradual GDA is a good contribution to this field.

**Claims And Evidence:**

Yes

**Claims Explanation:**

The paper supports its claims with both theoretical analysis and empirical results. The main theoretical arguments are clearly stated under explicit assumptions, and the experiments are generally consistent with the theoretical findings. After the rebuttal, the concerns raised by reviewers have been carefully addressed, which makes the overall evidence sufficiently convincing.